# BOUNDS OF CHAIN-OF-THOUGHT ROBUSTNESS: REASONING STEPS, EMBED NORMS, AND BEYOND

**Dingzirui Wang**[1*]  **Xuanliang Zhang**[1]   **Keyan Xu**[1]   **Qingfu Zhu**[1]
**Wanxiang Che**[1†]  **Yang Deng**[2]
[1]Harbin Institute of Technology    [2]Singapore Management University
`{dzrwang, xuanliangzhang, kyxu, qfzhu, car}@ir.hit.edu.cn`
`ydeng@smu.edu.sg`

## ABSTRACT

Existing research indicates that the output of Chain-of-Thought (CoT) is significantly affected by input perturbations. Although many methods aim to mitigate such impact by optimizing prompts, a theoretical explanation of how these perturbations influence CoT outputs remains an open area of research. This gap limits our in-depth understanding of how input perturbations propagate during the reasoning process and hinders further improvements in prompt optimization methods. Therefore, in this paper, we theoretically analyze the effect of input perturbations on the fluctuation of CoT outputs. We first derive an upper bound for input perturbations under the condition that the output fluctuation is within an acceptable range, based on which we prove that: *(i)* This upper bound is positively correlated with the number of reasoning steps in the CoT; *(ii)* Even an infinitely long reasoning process cannot eliminate the impact of input perturbations. We then apply these conclusions to the Linear Self-Attention (LSA) model, which can be viewed as a simplified version of the Transformer. For the LSA model, we prove that the upper bound for input perturbation is negatively correlated with the norms of the input embedding and hidden state vectors. To validate this theoretical analysis, we conduct experiments on three mainstream datasets and four mainstream models. The experimental results align with our theoretical analysis, empirically demonstrating the correctness of our findings.

## 1 INTRODUCTION

Chain-of-Thought (CoT) is an effective method that enhances the performance of large language models (LLMs) by prompting the model to generate a step-by-step reasoning process, thereby improving the quality of the results (Wei et al., 2022). However, numerous studies have indicated that CoT is highly sensitive to input, where subtle input perturbations can lead to significant performance fluctuations (Zhao et al., 2024; Shi et al., 2024b). To address this issue, researchers have proposed prompt optimization methods to enhance the reasoning performance of LLMs by refining the input prompt, lowering the effect of the input perturbation (Vatsal & Dubey, 2024; Sahoo et al., 2025). For instance, TextGrad (Yuksekgonul et al., 2025) optimizes prompts by constructing textual gradients, while OPRO (Yang et al., 2024) utilizes the LLM itself to iteratively generate more suitable prompts.

Despite this progress, a key gap remains: most studies treat CoT robustness as an empirical phenomenon, with little theoretical understanding of *why* and *how* perturbations propagate through the reasoning process of LLMs, thereby affecting output fluctuation. Without such analysis, our understanding of CoT robustness remains incomplete, and prompt optimization risks being limited to ad-hoc techniques. This motivates a fundamental research question: **what governs the CoT robustness of LLMs to input perturbations?**

Following the previous work (Huang et al., 2025), we consider CoT as a multistep iterative process, with the output of each step serving as the input for the next. Our theoretical analysis shows that

---

*Work was done during a visit at SMU.
†Corresponding Author.

Table 1: The main findings and corresponding evidence and experiment of this paper.

| Finding | Evidence | Experiment |
|---|---|---|
| More reasoning steps can reduce the effect of input perturbations. | Theorem 1 | §4.3 |
| The effect of input perturbations cannot be entirely eliminated by continuously increasing the number of CoT reasoning steps. | Equation 4 | §4.3 |
| CoT robustness is negatively correlated with the norms of the input embedding and hidden state vectors. | Theorem 2 | §4.4 |

under the assumption of Lipschitz continuity (Qi et al., 2023; Collins et al., 2025), longer CoT reasoning indeed reduces the fluctuation of outputs to input perturbations, but it never fully eliminates them. Even with an infinite number of CoT steps, a non-zero robustness bound remains, suggesting that CoT inherently dampens but cannot completely neutralize perturbations.

To further ground our analysis, we investigate robustness in the Linear Self-Attention (LSA) model (Wang et al., 2020a; Zhang et al., 2024a), which is commonly adopted as a simplified version of Transformer (Vaswani et al., 2017) for analysis without loss of generality. We prove that CoT robustness highly depends on model-level factors: the sensitivity to perturbations correlates negatively with the norm of the input vector and the hidden state vectors. Additionally, we discuss the impact of other factors in LSA on CoT robustness.

Finally, we validate our theory with experiments on four mainstream LLMs (Llama2, Llama3.1, Deepseek-R1-Distilled-Llama3.1, Qwen3) across three widely used reasoning datasets (MATH, MMLU-Pro, GPQA). The experimental results indicate that the variation in output fluctuation is consistent with the trends of the various factors identified in our theoretical analysis, thereby validating our findings. Furthermore, based on the analysis, we propose selecting the prompt by maximizing the upper bound of the input perturbation, which achieves consistent improvements over prior work, aiming to inspire future research in this area.

The main findings of our work are summarized in Table 1, and our main contributions are as follows:

- We provide an upper bound for the output fluctuation with respect to input perturbations under the assumption of Lipschitz continuity and prove that even an infinitely long CoT cannot completely counteract the impact of input perturbations.
- Taking the LSA model as a case study, we demonstrate that robustness to input perturbations is negatively correlated with the norms of the input and hidden state embedding vectors.
- Our experiments across multiple mainstream datasets and LLMs validate our theoretical analysis, and improvements based on our analysis also enhance performance compared to existing prompt optimization methods.

## 2 ROBUSTNESS OF CHAIN-OF-THOUGHT

In this section, we discuss the impact of input perturbations on the model output. We begin by providing some fundamental definitions. Then, we derive the upper bound for the output fluctuation given the input perturbation when the model satisfies Lipschitz continuity. Afterward, we determine the upper bound for the input perturbation when the output fluctuation is within an acceptable range. All the proof of this section is shown in Appendix C.

### 2.1 PRELIMINARY

Let $x, y \in \mathbb{R}^d$ denote the embedding vectors of the user query and the corresponding output, where $d$ is the dimension of the embedding space. Following previous works (Zhang et al., 2024a; Von Oswald et al., 2023), we directly use embedding vector instead of token embedding for analysis, as we consider the impact of user query as a whole on CoT robustness. We also prove that the conclusion is same with considering multiple tokens in Appendix D.5.

Let $\delta \in \mathbb{R}^d$ represent the input perturbation, and $\tilde{x} = x + \delta$ represent the perturbed input. Following previous work (Huang et al., 2025), we model the CoT reasoning process as a multistep iterative

procedure, where the output of each step serves as the input for the next step. Let $K \in \mathbb{N}^+$ be the total number of CoT reasoning steps, and let $h_{k,x} \in \mathbb{R}^d$ denote the hidden state at step $k$ taking $x$ as input[1]. Let $f(h,x) : \mathbb{R}^d \times \mathbb{R}^d \rightarrow \mathbb{R}^d$ represent the mapping function to generate the hidden state corresponding to an arbitrary reasoning model. Thus, we have $h_{1,x} = f(0,x)$ and $h_{k,x} = f(h_{k-1,x}, x)$. We denote the output fluctuation caused by the perturbation $\delta$ on step $k$ as $\varepsilon_k = h_{k,\tilde{x}} - h_{k,x}$.

## 2.2 Upper Bound of output fluctuation

We primarily discuss the impact of input perturbations on the model output under the assumption of Lipschitz continuity. Lipschitz continuity imposes a constraint on the growth rate of the model output, preventing an explosive increase. Considering that the output of current LLMs typically exhibits stable changes, many analytical works adopt this condition as a fundamental assumption (Qi et al., 2023; Collins et al., 2025). Specifically, for a given bivariate function $f(h,x)$, Lipschitz continuity requires the existence of constants $C, \gamma \in \mathbb{R}$ such that:

$$\|f(h_1, x_1) - f(h_2, x_2)\| \leq \gamma \|h_1 - h_2\| + C\|x_1 - x_2\| \tag{1}$$

Considering that the input $h$ at each step is the output of the previous step, by substituting it and expanding recursively, we can derive the following theorem:

**Theorem 1.** *If $f$ is Lipschitz continuous with respect to constants $C, \gamma \in \mathbb{R}$ as defined in Equation 1, then for a given input perturbation $\delta \in \mathbb{R}^d$, the upper bound of the corresponding output fluctuation $\varepsilon_K$ of the final step $K$ satisfies that:*

$$\|\varepsilon_K\| \leq \left( A\gamma^K + \frac{C}{1-\gamma}(1 - \gamma^K) \right) \|\delta\|$$

*where $A = \max \frac{\|\varepsilon_1\|}{\|\delta\|}$.*

From Theorem 1, we can observe that the propagation of the input perturbation can be mainly divided into two parts. *(i)* The part contained in the hidden state vector: since the hidden state vector is updated at each step, the coefficient of this part of the perturbation is continuously multiplied by the corresponding Lipschitz constant $\gamma$; *(ii)* The part contained in the input vector: since the input vector at each step does not change, this part of the perturbation gradually accumulates at each step, and thus the corresponding perturbation coefficient is $\sum_{k=1}^{K} C\gamma^k = \frac{C}{1-\gamma}(1 - \gamma^K)$.

Besides, considering that when the model is fixed, the corresponding parameters $C$ and $\gamma$ are also fixed. Therefore, based on Theorem 1, the upper bound of the output fluctuation primarily depends on two factors: the number of reasoning steps $K$ and the magnitude of the perturbation $\|\delta\|$. Based on previous work (Zhou et al., 2020; Diehl Martinez et al., 2024), we assume that $\gamma < 1$, which implies that for a well-trained model, the output fluctuation gradually converges to a fixed value rather than diverging infinitely. We also fit the values of $\gamma$ in Appendix F.1 using practical datasets and models. Consequently, as the number of reasoning steps $K$ increases, the corresponding upper bound of output fluctuation decreases, indicating that the increment of CoT steps can mitigate the impact of input perturbations on the model output.

## 2.3 Upper Bound of Input Perturbation

In practical applications, a model can tolerate a certain degree of output fluctuation while maintaining the same final result. For example, in a classification task, as long as the probability of the same option remains the highest before and after the input perturbation, a certain level of fluctuation in the output probabilities can not affect the final answer. Therefore, we assume there exists an acceptable boundary $R \in \mathbb{R}^+$, such that we consider the output fluctuation to be acceptable when the following condition is met:

$$\|\varepsilon\| \leq R \tag{2}$$

---

[1]In practical models, $x$ can be viewed as the input embedding vector, and $h$ can be viewed as the encoded vector from the last layer of the model.

To ensure that the norm of output fluctuation $\varepsilon$ is less than $R$, we require the expression on the right-hand side of the inequality in Theorem 1 to be less than $R$, which yields:

$$\|\delta\| \leq \frac{R}{A\gamma^K + \frac{C}{1-\gamma}(1-\gamma^K)} \tag{3}$$

It can be observed that the upper bound of the input perturbation is mainly influenced by $R, C$, and $\gamma$. A larger $R$ indicates that a greater output fluctuation is acceptable, thus leading to a larger upper bound for the input perturbation. Conversely, larger values of $C$ and $\gamma$, according to Equation 1, suggest that the model is less capable of compressing the output fluctuation, implying a weaker ability to handle input perturbations, which results in a smaller upper bound for the input perturbation.

Taking $\gamma < 1$ and letting $K \rightarrow \infty$, we can obtain that:

$$\|\delta\| \leq \frac{R(1-\gamma)}{C} \tag{4}$$

Equation 4 indicates that the effect of extending the reasoning process to eliminate input perturbation is limited. Even with an infinitely long reasoning process, if the input perturbation exceeds a certain threshold, the model cannot eliminate the resulting output fluctuation. For example, if we "perturb" a numerical reasoning problem into a coding problem, the model cannot generate the answer to the original problem, regardless of the reasoning length.

## 3 CHAIN-OF-THOUGHT ROBUSTNESS ON LINEAR SELF-ATTENTION

According to the discussion in §2.3, the upper bound of input perturbation that a model can tolerate using CoT depends on the properties of the model itself. Therefore, in this section, we discuss the factors that influence the upper bound of input perturbation on the Linear Self-Attention model (LSA) (Wang et al., 2020a; Zhang et al., 2024a), which can be viewed as a simplified version of the current mainstream Transformer architecture. All the proofs of this section are shown in Appendix C. We also discuss the influence of various *non-linear factors* in the Transformer on the conclusions of this section in Appendix D.1, and we analyze none-linear attention in Appendix D.3.

### 3.1 DEFINITION OF LINEAR SELF-ATTENTION

We first define LSA following the previous work (Wang et al., 2020a; Zhang et al., 2024a). Let $W^{KQ}, W^{PV} \in \mathbb{R}^{b \times b}$ denote the combined query-key and projection-value matrices, and let $\rho \in \mathbb{R}^+$ be the normalization factor. We denote the parameters as $\theta = (W^{KQ}, W^{PV}, \rho)$. Let $E = [h, x]$. The LSA is then defined as:

$$f_{LSA}(h, x; \theta) = E + W^{PV}E\frac{E^\top W^{KQ}E}{\rho} \tag{5}$$

LSA can be viewed as replacing the non-linear softmax mapping in a single-layer Transformer with a linear mapping. Following prior work (Zhang et al., 2024a), we set $\rho = 1$ in this paper.

Based on Equation 5, Zhang et al. (2024a) proves that for a well-trained LSA on the training data $\{(h_i, x_i, y_i)_N\}$, its parameters $\theta$ must satisfy:

$$W_*^{KQ} = [\text{Tr}(\Gamma^{-2})]^{-\frac{1}{4}}\begin{pmatrix}\Gamma^{-1} & 0_d \\ 0_d^\top & 0\end{pmatrix}, W_*^{PV} = [\text{Tr}(\Gamma^{-2})]^{\frac{1}{4}}\begin{pmatrix}0_{d \times d} & 0_d \\ 0_d^T & 1\end{pmatrix} \tag{6}$$

where $\Gamma = \left(1 + \frac{1}{N}\right)\Lambda + \frac{1}{N}\text{Tr}(\Lambda)I_d \in \mathbb{R}^{d \times d}$ and $\Lambda$ denote the covariance matrix of the training data. Substituting these optimal parameters into the equation yields:

$$f_{LSA}(h, x; \theta_*) = E + \begin{pmatrix}0_{d \times d} & 0_d \\ 0_d^T & 1\end{pmatrix}E\frac{E^\top \begin{pmatrix}\Gamma^{-1} & 0_d \\ 0_d^\top & 0\end{pmatrix}E}{\rho} \tag{7}$$

Considering the gradient explosion without the residual flow, we introduce the residual coefficient $\eta \in (0, 1)$ to LSA (Zhang et al., 2019; Bachlechner et al., 2020). The corresponding function is:

$$f_{LSA}(h, x; \theta_*) = \eta E + \begin{pmatrix}0_{d \times d} & 0_d \\ 0_d^T & 1\end{pmatrix}E\frac{E^\top \begin{pmatrix}\Gamma^{-1} & 0_d \\ 0_d^\top & 0\end{pmatrix}E}{\rho} \tag{8}$$

Next, we use Equation 8 as the prediction function $f_{LSA}(h, x)$ to discuss the effect of input perturbations on the LSA output.

## 3.2 INPUT ROBUSTNESS OF LINEAR SELF-ATTENTION

Based on Equation 8, we provide the upper bounds for the two Lipschitz constants in Equation 1:

**Lemma 1.** *If $\|x\| \leq R_x$ and $\|h\| \leq R_h$, let $\alpha = (\text{Tr}(\Gamma^{-2}))^{-\frac{1}{4}}$. Then we have:*

$$C \leq \eta + \alpha^{-1} \|\Gamma^{-1}\| R_h^2$$

$$\gamma \leq \sqrt{\eta^2 + 4 R_x^2 \alpha^{-2} \|\Gamma^{-1}\|2^2 R_h^2}$$

The assumption of $R_x$ and $R_h$ in Lemma 1 bounds the norms of $x$ and $h$. Considering that excessively large embedding vectors can lead to unstable inference, the embedding vector norms in mainstream LLMs are typically confined within a certain range (Fazlyab et al., 2019; Kim et al., 2021), making this assumption reasonable.

By substituting the upper bounds of $C$ and $\gamma$ from Lemma 1 into the Equation 3, we can obtain that:

**Theorem 2.** *If $\|x\| \leq R_x$ and $\|h\| \leq R_h$ and let*

$$\alpha = \left[\text{Tr}(\Gamma^{-2})\right]^{\frac{1}{4}}, \qquad s = \|\Gamma^{-1}\|, \qquad \beta = \alpha^{-1} s R_h^2, \qquad \gamma = \sqrt{\eta^2 + 4 R_x^2 \alpha^{-2} s^2 R_h^2}.$$

*With $A > 0$ such that $\|e_0\| \leq A\|\delta\|$, the certified tolerable input-perturbation radius of the LSA map for the reasoning step $K \in \mathbb{N}^+$ is:*

$$\|\delta\| \leq \frac{(1 - \gamma) R}{(\eta + \beta) + (A(1 - \gamma)(1 + \beta)) \gamma^K}$$

*In particular, if $\gamma < 1$, as $K \to \infty$, we can derive that:*

$$\|\delta\| \leq \frac{(1 - \gamma) R}{\eta + \beta}$$

According to Theorem 2, the impact of input perturbations on model outputs primarily depends on:

- $R$: The range of output perturbation that is acceptable. *A larger range indicates a greater tolerance for perturbations*, leading to a higher upper bound for the input perturbation.

- $R_x$: The tolerable perturbation radius is negatively correlated with $R_x$, indicating that *a larger norm of the input lowers the model's robustness to input perturbations*. According to the proof, a larger $R_x$ leads to a larger coefficient of the perturbation in the resulting bound.

- $R_h$: The tolerable perturbation radius is negatively correlated with $R_h$. This suggests that *a larger norm of the internal state makes the model more susceptible to being led astray* during the reasoning process, thus weakening its resistance to input perturbations.

- $\Gamma$: The covariance matrix of the training data. *More inconsistent training data leads to the model being more sensitive to input perturbations*.

- $\eta$: A larger residual coefficient indicates that the model retains more information from input, causing *the effects of input perturbations to be preserved across layers*.

The theoretical results suggest two main robustness levers at inference and training time. At inference, Theorem 1 shows that longer, more structured chains of thought reduce output fluctuations, while smaller norms of input embeddings and encoded representations decrease the effective Lipschitz constants in the bounds. At training time, Theorem 2 indicates that reducing representation norms and decreasing the training-data covariance ($\Gamma$) (i.e., making the data more consistent), increases the certified perturbation radius.

We further discuss the impact of vector norms on the CoT robustness in Appendix D.4. Considering that verifying the effects of $\Gamma$ and $\eta$ requires modifying the training data and the model architecture of LLMs, this work provides only a theoretical analysis of $\Gamma$ and $\eta$ to inspire future work on corresponding empirical studies, while focusing on verifying the effects of $R$, $R_x$, and $R_h$.

Table 2: Average exact match (EM) and output fluctuation (OF) of different models using various prompts on different datasets. The highest EM and lowest OF under each setting is marked in **bold**.

| Model | MATH | | MMLU-Pro | | GPQA | |
|---|---|---|---|---|---|---|
| | EM | OF | EM | OF | EM | OF |
| Llama2-7b | $14.2 \pm 5.0$ | 0.475 | $11.2 \pm 5.7$ | 0.622 | $17.5 \pm 4.7$ | 0.509 |
| Llama3.1-8b | $45.8 \pm 7.2$ | 0.366 | $41.0 \pm 10.7$ | 0.350 | $26.6 \pm 5.7$ | 0.467 |
| Llama-R1-8b | $64.8 \pm 3.0$ | 0.158 | $44.8 \pm 8.3$ | 0.292 | $28.5 \pm 2.9$ | 0.371 |
| Qwen3-8b | $\mathbf{77.2 \pm 1.6}$ | **0.097** | $\mathbf{46.9 \pm 5.2}$ | **0.162** | $\mathbf{37.3 \pm 1.9}$ | **0.214** |

## 4 EXPERIMENT

### 4.1 EXPERIMENT SETUP

**Dataset** Our experiments are conducted on three reasoning datasets: MATH (Hendrycks et al., 2021), MMLU-Pro (Wang et al., 2024c), and GPQA (Rein et al., 2024). Detailed descriptions of these three datasets are provided in Appendix E.1. Considering the high difficulty of these datasets, which require multistep reasoning processes for solutions, we suppose they can effectively reflect the influence of various factors on the model's ability to handle input perturbations. We also adapt experiments on more datasets in Appendix F.2.

**Model** We conduct experiments on four mainstream LLMs including: Llama2-7b (Touvron et al., 2023), Llama3.1-8b (Grattafiori et al., 2024), Deepseek-R1-Distilled-Llama3.1-8b (Llama-R1-8b) (DeepSeek-AI et al., 2025) and Qwen3-8b (Yang et al., 2025). These models cover a range of capabilities, allowing for a comprehensive evaluation of how model type and different factors affect the handling of input perturbations. For Llama2-7b and Llama3.1-8b, we employ the instruct version. For Qwen3-8b, we utilize its *Thinking Mode* to fully leverage its performance. We also experiment with the performance under different model scales in Appendix F.5.

**Metric** To evaluate both the performance and robustness, we adopt the following two metrics: ***(i)*** ***Exact Match (EM)***: Whether the predicted answer is the same as the correct answer to the question. A higher value for this metric indicates that the model is better at solving the given dataset, reflecting the overall performance in a specific setting. ***(ii)*** ***Output Fluctuation (OF)***: The normalized entropy (Friedrich, 2021) of the answers generated from different prompts for the question. A higher value for this metric indicates that the output on the given question is less consistent, reflecting the robustness of the specific setting. We detail how to calculate OF in Appendix E.4. We also evaluate with the other fluctuation metric in Appendix F.3.

**Input Perturbations** To fully reflect the robustness to input perturbations, for each model and dataset, we first generate multiple prompts. Then, for each question, we use these prompts to generate multiple answers. We evaluate the performance by analyzing the correctness and consistency of these answers. To ensure the reliability of our results, we collect all prompts during the optimization of three mainstream methods, including TextGrad (Yuksekgonul et al., 2025), OPRO (Yang et al., 2024), and CFPO (Liu et al., 2025). The base prompts used follow Grattafiori et al. (2024), which is shown in Appendix E.2. The number of prompts used for each dataset and model is detailed in Appendix E.3. We also adapt the experiments using the same prompts on different datasets and models in Appendix F.4.

More experimental setups are detailed in Appendix E.5.

### 4.2 OVERALL EVALUATION

**CoT Robustness Scales with Model Capability**   The average performance and corresponding fluctuations for different prompts across various datasets and models are shown in Table 2. Results show that across all models, as their capabilities increase, not only does the average EM improve, but the corresponding output fluctuation also decreases. Regarding different tasks, multiple-choice sets (MMLU-Pro, GPQA) exhibit larger fluctuation than MATH, where small logit shifts can flip the selected option (Pezeshkpour & Hruschka, 2024; Wang et al., 2024a). Yet on GPQA, despite lower EM, fluctuation is not excessive, suggesting *difficulty* alone does not significantly affect the CoT robustness. Interpreted through our bounds, stronger models typically (i) train on *data with higher consistency* (better cleaning and synthesis) which increases the upper bound of input perturbations,

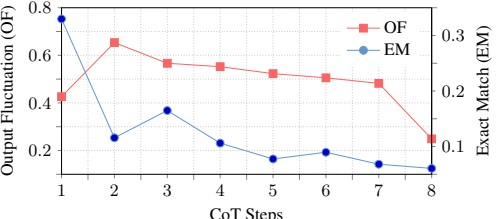

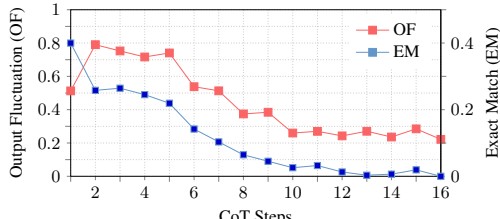

Figure 2: The change in OF (left Y-axis) and EM (right Y-axis) with the reasoning steps of the generated CoT, averaged across all experimental datasets and models. The X-axis denotes the CoT step and Y-axis denotes the value of each metric. The curves at X and Y axes illustrate the data distribution. The CoT steps are segmented using ROSCOE (Golovneva et al., 2023).

Figure 3: The change in OF (left Y-axis) and EM (right Y-axis) with the reasoning steps on all datasets and models under the reasoning steps from 1 to 16. The X-axis denotes the CoT step and Y-axis denotes each metric. The curves at X and Y axes illustrate the data distribution. The CoT steps are segmented using ROSCOE (Golovneva et al., 2023).

which is governed by the data-consistency constant $\Gamma$ in Theorem 2, and (ii) yield *longer, more structured reasoning steps*, increasing $K$ in Theorem 1 and thereby tightening the fluctuation bound. Models supporting Long-CoT (Chen et al., 2025) (e.g., Llama-R1, Qwen3) exemplify this effect. Notably, some settings exhibit larger fluctuations in EM despite having smaller OF. This occurs because the average EM differs across settings, where a setting with a high average EM, even a minor output fluctuation can result in a large absolute EM fluctuation. In contrast, OF directly measures the consistency of the outputs, thus offering a more faithful representation of output robustness.

**Greater Input Perturbation Makes Output Less Robust** To observe the effect of input perturbation on the model output, we analyze the change in output fluctuation with respect to the input perturbation across all datasets and models. For each question, the input perturbation is calculated as the average distance of input embedding vectors from their mean vector. The results are shown in Figure 1. From the figure, we can find that: *(i)* As the input perturbation increases, the output fluctuation also increases (Pearson Correlation Coefficient = 0.619), which supports the conclusion of Theorem 1. *(ii)* The majority of input perturbations are concentrated in the range of less than 0.1, yet the corresponding change in output fluctuation is quite large, which indicates that even slight fluctuations in the input can lead

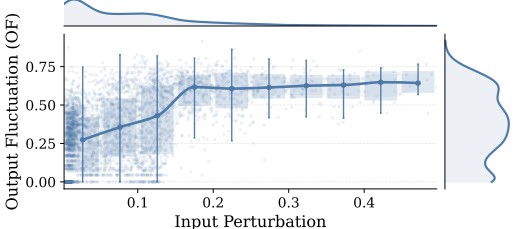

Figure 1: The output fluctuation across input perturbation on all datasets and models. Each point denotes one question, where X-axis denotes the input perturbation as the average distance of embedding vectors from their mean vector, and Y-axis denotes OF. The curves at X and Y axes illustrate the data distribution. The Pearson correlation coefficient is 0.619.

to significant fluctuations in the output, which is consistent with the findings of previous studies (Zhao et al., 2024; Bigelow et al., 2024). *(iii)* When the input perturbation exceeds 0.2, the output fluctuation becomes relatively robust as the input perturbation increases. This is because the output fluctuation is measured using normalized entropy, whose maximum possible value depends on how many prompts are used to generate answers. This means that even when input changes become larger, the maximum possible fluctuation in the output stays roughly constant.

## 4.3 IMPACT OF REASONING STEP LENGTH ON COT ROBUSTNESS

**Robustness is Positively Correlated with Reasoning Step Length** To verify the impact of reasoning steps, we analyze performance as a function of CoT steps (steps computed following ROSCOE (Golovneva et al., 2023)). The experimental results are presented in Figure 2. Figure 2 reveals the following trends: *(i)* Output fluctuation generally *decreases* as steps increase, matching Theorem 1: larger $K$ tightens the robustness bound. *(ii)* The model output fluctuation is relatively low for one-step CoT cases because trivially solvable items need little reasoning and are stable even with short chains. *(iii)* The performance (*i.e.*, EM) does *not* not necessarily increase with $K$. More steps often correlate with harder items, so accuracy can drop as $K$ rises.

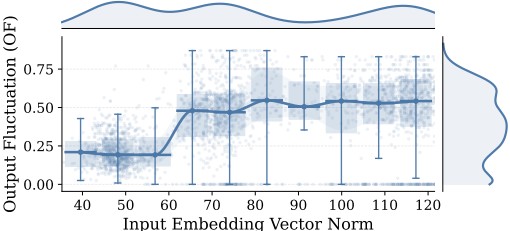 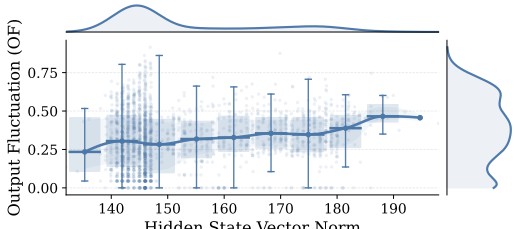

Figure 4: The change in output fluctuation with the norm of the input embedding vector across all experimental datasets and models. Each point denotes the result of one question, where X-axis denotes the input vector norm and Y-axis denotes OF of this question. The Pearson correlation coefficient is 0.506.

Figure 5: The change in output fluctuation with the norm of the hidden state vector across all experimental datasets and models. Each point denotes the result of one question, where X-axis denotes the hidden state vector norm and Y-axis denotes OF of this question. The Pearson correlation coefficient is 0.229.

**Infinity Reasoning Steps Cannot Eliminate the Impact of Input Perturbation**  To further verify the conclusion from Equation 4 that even infinitely long reasoning steps cannot completely eliminate the impact of input perturbations, we conduct experiments with an extended number of reasoning steps. We add the instruction "`You MUST reason in exactly` $K$ `steps`" to the prompt to guide the model in generating longer reasoning processes, requiring the model to generate outputs for $K = 1, ..., 16$ steps. Considering that the model could not strictly follow the instruction to generate the specified steps, we still use ROSCOE to calculate the actual steps. The results are presented in Figure 3. From the figure, we observe that as the number of reasoning steps increases, the output fluctuation decreases but eventually converges to a relatively stable level. This indicates that the role of reasoning steps in eliminating perturbations is limited, thereby empirically validating the conclusion of Equation 4. Since OF begins to fluctuate, we suppose that current experimental steps have supported our conclusion and do not conduct experiments over 16 steps.

## 4.4 IMPACT OF EMBEDDING NORMS ON COT ROBUSTNESS

**Larger Input Embedding Norm Makes Output Less Robust**  To verify the relationship between output fluctuation and the norm of the input embedding vector, we analyze the experimental results across all datasets and models, as shown in Figure 4. From the figure, we can observe that: *(i)* As the norm of the input embedding vector increases, the model output fluctuation shows a general upward trend, which confirms the related conclusions in Theorem 2. *(ii)* As the input embedding norm grows, output fluctuation saturates, since a normalized entropy capped by the number of prompts, its maximum stays roughly constant even under larger input perturbations. *(iii)* When the norm of the input embedding vector increases from 60 to 70, the output fluctuation exhibits a sudden jump, which indicates that a threshold exists for the vector norm that the model can handle stably. Once this threshold is surpassed, most input perturbations exceed the upper bound defined in Theorem 2, causing significant fluctuations in the output.

**Larger Hidden State Norm Makes Output Less Robust**  To verify the relationship between the norm of the hidden state vector and the output fluctuation, we analyze the results across all datasets and models. The hidden state vector is extracted from the last layer of the last CoT step. The results are shown in Figure 5. From the figure, we can find that: *(i)* As the norm of the hidden state increases, the output fluctuation shows a general upward trend, which confirms the positive correlation between the two as stated in Theorem 2. *(ii)* The vector norms for the majority of data points are concentrated on the $(140, 150)$ range, which indicates that a well-trained model tends to encode data into a specific and relatively small norm interval to mitigate the impact of input perturbations. *(iii)* Overall, the change in output fluctuation with the hidden state norm is not significant. We suppose the reasons for this are that the constant $\gamma$ is determined by the upper bound of the hidden state norm rather than its specific value, and that the various normalization structures like LayerNorm (Xiong et al., 2020) within the Transformer architecture mitigate the output fluctuation to some extent.

## 4.5 PROMPT OPTIMIZATION WITH HIGHER INPUT ROBUSTNESS

To shed light on future research, we discuss how to optimize the performance of prompt optimization based on Theorem 2. Let $\tau = \alpha^{-1}s$ and $F$ denote the expression on the right-hand side of Theorem 2. We hope to select the prompt that makes $F$ as large as possible, thereby increasing the upper bound of the input perturbation. Let $A = (R_x R_h)^2$, we can derive that:

$$\frac{\partial F}{\partial A} = -\frac{R\tau^2}{2(\eta + \tau R_h^2)\sqrt{\eta^2 + \tau^2 A}} < 0 \quad (9)$$

This shows that $F$ is negatively correlated with $A$, where a larger value of $A^{-1}$ corresponds to a larger upper bound for the input perturbation, meaning the model can tolerate greater input perturbations. Therefore, for each question, we first

Table 3: EM on each model and dataset using different prompt optimization methods. "-" denotes using the single base prompt directly. The best result under each setting is marked in **bold**.

| Model | Method | MATH | MMLU-Pro | GPQA |
|---|---|---|---|---|
| Llama3.1-8b | - | 46.8 | 45.7 | 23.7 |
| | TextGrad | 45.2 | 47.4 | 27.6 |
| | OPRO | 44.6 | 47.1 | 27.1 |
| | CFPO | 47.0 | 48.1 | 27.6 |
| | Ours | **47.2** | **49.0** | **32.3** |
| Qwen3-8b | - | 77.4 | 42.3 | 37.1 |
| | TextGrad | 77.6 | 44.9 | 38.4 |
| | OPRO | 77.2 | 45.9 | 37.4 |
| | CFPO | 77.0 | 45.8 | 38.4 |
| | Ours | **77.6** | **49.2** | **38.4** |

construct inputs using all obtained prompts and extract the corresponding embedding layer vectors, as well as the vectors from the final layer to serve as hidden state vectors. We then calculate the norms of both to obtain $A$. Subsequently, for each question, we select the prompt with the highest value of $A^{-1}$ as the designated prompt for inference. The experimental results are shown in Table 3. We calculate only the Exact Match (EM) for each method and not the Output Fluctuation (OF), since each method selects a single optimal prompt for each question to perform inference, and consequently yields only one output as the final answer. From the table, we can see that our method brings performance improvements across all settings, which demonstrates the effectiveness of our method. We also discuss the efficiency of our method in Appendix D.2. Since the primary objective of this paper is to analyze the factors affecting input robustness, rather than to optimize prompt optimization methods, we leave the investigation into how to better effectiveness and efficiency, and a more extensive comparison with additional baselines for future work.

## 5 RELATED WORKS

**Robustness of Chain-of-Thought.** Numerous studies show that slight perturbations in the input can lead to drastic changes in the output of CoT (Zhao et al., 2024; Shi et al., 2024b). Therefore, to enhance the performance of CoT, a variety of works are proposed to improve and analyze the CoT robustness. For example, noisy or off-task rationales reliably degrade CoT performance. Contrastive denoising, including CD-CoT and NoRa, mitigates these effects (Zhou et al., 2024). Break-The-Chain applies semantics-preserving rewrites (narrativization, mild constraint changes, reordering, numeric tweaks) to reveal sensitivity in code generation (Roh et al., 2025). Character-level perturbations ($R^2$ATA) likewise disrupt reasoning (Gan et al., 2024). Chain-of-Defensive-Thought structures defensive rationales that resist corruption or injection and reduce collapse (Wang et al., 2025). Post-hoc Self-Correction Reflection repairs errors under perturbations (Wu et al., 2025). Self-Consistency reduces single-path brittleness through voting (Wang et al., 2023b). CoT is sensitive to step order and exemplar relevance (Wang et al., 2023a). Theory indicates that more coherent chains aid error correction but increase vulnerability to noise in intermediate steps (Cui et al., 2024). Evidence also suggests that CoT often functions as constrained imitation rather than genuine reasoning (Shao & Cheng, 2025). Generalization analyses for nonlinear Transformers identify robustness conditions under noise and distribution shift (Li et al., 2024).

Despite these advances, the mechanism by which input perturbations induce output changes remains under-specified. We derive upper bounds that link input perturbations to output fluctuations and analyze the factors that govern CoT robustness, extending prior research.

**Prompt Optimization.** Prompt optimization methods primarily focus on how to optimize prompts based on the given model and task to enhance the performance. Work on prompt optimization spans RL and gradient-free edit search (Deng et al., 2022; Prasad et al., 2023), influential-token clustering to shrink the search space (Zhou et al., 2023a), and ensemble-style boosting to avoid single-prompt failure (Hou et al., 2023). Refinements include genetic and actor–critic editing, localized zeroth-order updates, and exemplar-ordering optimization (Xu et al., 2022; Dong et al., 2024; Hu et al., 2024; Wu et al., 2024). APE and OPRO iteratively propose and select improved instructions (Zhou

et al., 2023b; Yang et al., 2024). ProTeGi and APO implement textual "gradient descent" with beam or bandit search (Pryzant et al., 2023). TextGrad generalizes to "automatic differentiation via text" (Yuksekgonul et al., 2025). Data-driven pipelines such as Self-Instruct and Auto-Instruct bootstrap and rank prompt sets (Wang et al., 2023c; Zhang et al., 2023). Search strategies include MCTS with reflective error analysis (Wang et al., 2024b). Budgeted best-arm identification supports selection under tight evaluation budgets (Shi et al., 2024a). Preference-based black-box optimization aligns prompts with user goals (Cheng et al., 2024). RL improves textual-prompt stability (Kwon et al., 2024). Compiler-style systems such as DSPy learn prompts for multi-stage LM pipelines (Khattab et al., 2024). OPRO-like gains may attenuate on smaller open models (Zhang et al., 2024b).

Despite strong empirical progress, the mechanism pathway from input perturbations to output fluctuations remains poorly understood. We analyze this affect and its determinants to guide principled designs for the future prompt optimization works.

## 6 CONCLUSION

In this paper, we theoretically analyze the influence of various factors on the input robustness of CoT. We first prove that the impact of input perturbations on the CoT output is negatively correlated with the number of CoT reasoning steps, and that even an infinite number of steps cannot completely eliminate the effects of input perturbations. We then apply these findings to LSA, demonstrating that its input robustness is negatively correlated with the norms of the input embedding and hidden state vectors. To validate these conclusions, we conduct experiments on four mainstream LLMs and three mainstream datasets. Experimental results reveal that output fluctuations vary with different factors in line with our expectations, supporting the validity of our findings. Furthermore, guided by this analysis, we propose to select the prompt by raising the upper bound of input perturbation, which yields consistent performance gains over previous works. Moving forward, our work opens several promising avenues for advancing robust chain-of-thought reasoning. In particular, a key next step is to systematically examine how the parameters $\Gamma$ and $\eta$ in Theorem 2 influence input robustness, which could also inform the design of more resilient large reasoning models.

## 7 REPRODUCIBILITY

We have provided all proofs of this paper in Appendix C. The experimental and pre-processed data and code are released in `https://github.com/zirui-HIT/CoT-Robust`.

## ACKNOWLEDGE

We gratefully acknowledge the support of the National Natural Science Foundation of China (NSFC) via grant 62236004 and 62476073. This research/project was also supported by the Singapore Ministry of Education (MOE) Academic Research Fund (AcRF) Tier 1 grant (Proposal ID: 24-SIS-SMU-002). Yang Deng is support by the Lee Kong Chian Fellowship awarded by Singapore Management University.

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

## A    ETHICS STATEMENT

All datasets and models used in this paper are publicly available, and our usage follows their licenses and terms.

## B    LLM USAGE

We have employed the AI tool for coding and writing polishing.

## C    PROOFS

*Proof of Theorem 1.*

$$\varepsilon_k := h_k(x+\delta) - h_k(x), \qquad k \in \mathbb{N}^+.$$

$$h_k(x) = f(h_{k-1}(x), x), \qquad h_k(x+\delta) = f\big(h_{k-1}(x+\delta), x+\delta\big).$$

$$\|\varepsilon_k\| = \big\|f(h_{k-1}(x+\delta), x+\delta) - f(h_{k-1}(x), x)\big\|.$$

$$\|f(h_1, x_1) - f(h, x)\| \le \gamma\|h_1 - h\| + C\|x_1 - x\|.$$

$$\Rightarrow \quad \|\varepsilon_k\| \le \gamma\|\varepsilon_{k-1}\| + C\|\delta\|.$$

$$\Rightarrow \quad \|\varepsilon_K\| \le \gamma^K\|\varepsilon_0\| + C\|\delta\| \sum_{i=0}^{K-1} \gamma^i.$$

$$\sum_{i=0}^{K-1} \gamma^i = \frac{1-\gamma^K}{1-\gamma} \quad (\gamma \in [0,1)).$$

$$\Rightarrow \quad \|\varepsilon_K\| \le \gamma^K\|\varepsilon_1\| + \frac{C}{1-\gamma}(1-\gamma^K)\|\delta\|.$$

$$A := \max \frac{\|\varepsilon_1\|}{\|\delta\|}.$$

$$\Rightarrow \quad \boxed{\|\varepsilon_K\| \le \left(A\gamma^K + \frac{C}{1-\gamma}(1-\gamma^K)\right)\|\delta\|}$$

$\square$

*Proof of Lemma 1.*

$$E = \begin{bmatrix} h \\ x \end{bmatrix}, \qquad f(E) = \eta E + (PE)\, s(E), \qquad s(E) = E^\top K E.$$

$$P = W_*^{PV} = [\mathrm{Tr}(\Gamma^{-2})]^{\frac{1}{4}} \begin{bmatrix} 0 & 0 \\ 0 & 1 \end{bmatrix}, \qquad K = W_*^{KQ} = [\mathrm{Tr}(\Gamma^{-2})]^{-\frac{1}{4}} \begin{bmatrix} \Gamma^{-1} & 0 \\ 0 & 0 \end{bmatrix}.$$

$$K_s := \tfrac{1}{2}(K + K^\top) = K, \qquad \nabla s(E) = 2K_s E.$$

$$\nabla f(E) = \eta I + s(E)P + (PE)\,(2K_s E)^\top. \tag{$\star$}$$

**Bound for $C$.**

$$\frac{\partial f}{\partial x}(E) = \eta\,\Pi_x + s(E)\,P_x + (PE)\,\big(2K_s E\big)_x^\top.$$

$$K = \begin{bmatrix} * & 0 \\ 0 & 0 \end{bmatrix} \;\Rightarrow\; (K_s E)_x = 0 \;\Rightarrow\; (2K_s E)_x = 0.$$

$$\Rightarrow \quad \frac{\partial f}{\partial x}(E) = \eta\,\Pi_x + s(E)\,P_x.$$

$$\left\|\frac{\partial f}{\partial x}(E)\right\| \le \eta + \|P_x\|\,|s(E)| \le \eta + \|P\|\,\|K\|\,\|E_h\|^2.$$

$$E_h = \begin{bmatrix} h \\ 0 \end{bmatrix}, \quad \|E_h\| = \|h\| \le R_h, \quad \|P\| = [\mathrm{Tr}(\Gamma^{-2})]^{\frac{1}{4}}, \quad \|K\| = [\mathrm{Tr}(\Gamma^{-2})]^{-\frac{1}{4}}\|\Gamma^{-1}\|.$$

$$\Rightarrow \quad \boxed{C \;\le\; \eta + \|\Gamma^{-1}\|\,R_h^2} \quad (\text{i.e., } C \le \eta + [\mathrm{Tr}(\Gamma^{-2})]^{-\frac{1}{4}}\|\Gamma^{-1}\|\,R_h^2).$$

**Bound for $\gamma$.**

$$\frac{\partial f}{\partial h}(E) = \eta\,\Pi_h + s(E)\,P_h + (PE)\left(2K_s E\right)_h^\top.$$

$$P_h = 0 \quad (\text{since } PE = [0;\, [\text{Tr}(\Gamma^{-2})]^{\frac{1}{4}}x]),$$

$$(2K_s E)_h = 2\,[\text{Tr}(\Gamma^{-2})]^{-\frac{1}{4}}\Gamma^{-1}h.$$

For any $v \in \mathbb{R}^d$,

$$\frac{\partial f}{\partial h}(E)v = \begin{bmatrix} \eta v \\ 0 \end{bmatrix} + \begin{bmatrix} 0 \\ \|PE\| \cdot \frac{2\,\|\Gamma^{-1}h\|}{[\text{Tr}(\Gamma^{-2})]^{\frac{1}{4}}}\,\frac{(h^\top\Gamma^{-1}v)}{\|\Gamma^{-1}h\|}\,\frac{PE}{\|PE\|} \end{bmatrix}.$$

$$\|PE\| = [\text{Tr}(\Gamma^{-2})]^{\frac{1}{4}}\|x\| \le [\text{Tr}(\Gamma^{-2})]^{\frac{1}{4}}R_x, \qquad \|\Gamma^{-1}h\| \le \|\Gamma^{-1}\|\,\|h\| \le \|\Gamma^{-1}\|R_h.$$

$$(\text{orthogonal blocks}) \quad \Rightarrow \quad \left\|\frac{\partial f}{\partial h}(E)v\right\|^2 \le \eta^2\|v\|^2 + \left(2\,R_x\,\|\Gamma^{-1}\|\,R_h\right)^2\|v\|^2.$$

$$\Rightarrow \quad \boxed{\gamma \le \sqrt{\eta^2 + 4\,R_x^2\,\|\Gamma^{-1}\|^2\,R_h^2}} \quad (\text{i.e., } \gamma \le \sqrt{\eta^2 + 4\,R_x^2\,[\text{Tr}(\Gamma^{-2})]^{-\frac{1}{2}}\,\|\Gamma^{-1}\|^2\,R_h^2}).$$

$\square$

# D  ADDITIONAL DISCUSSION

## D.1  INFLUENCE OF NON-LINEAR FACTORS OF TRANSFORMER

In this section, we discuss the influence of different non-linear factors within the Transformer architecture on the conclusions of Theorem 2. Overall, most non-linear factors contribute to enhancing the model's input robustness. Due to the complexity of theoretically proving the effects of these non-linear factors, we only provide an intuitive analysis and leave rigorous mathematical proofs for future work.

**Attention Non-linearity (Softmax)**  The exponential normalization of Softmax produces sharp distributions at low temperatures or with large logit scaling, leading to a "winner-takes-all" switching behavior among highly competitive keys. Intuitively, this amplifies the sensitivity to perturbations in the input and intermediate states, which is equivalent to increasing the effective Lipschitz constant ($\gamma$) and the input channel coefficient ($C$). It also causes locally quasi-discrete transitions in attention weights. Therefore, within the framework of Theorem 2, sharper attention typically reduces the tolerable perturbation radius. Conversely, smoother attention (achieved with high temperature or small scaling factors) mitigates this sensitivity, thereby increasing the robustness radius.

**Non-linearity of Normalization Layers (LayerNorm/RMSNorm)**  Normalization explicitly constrains the norm of hidden states through demeaning and scaling by variance. When statistics are stable, this effectively suppresses $R_h$ and weakens the amplification chain across layers, manifesting as smaller effective values for $\gamma$ and $C$. This aligns with the monotonic relationship described in Theorem 2, where a smaller norm corresponds to stronger robustness. However, it is important to note that when the intra-layer variance becomes abnormally small (close to zero), the scaling factor can locally amplify noise, creating transient high-gain regions and leading to edge cases where robustness decreases. Therefore, stable statistics and moderate pre-scaling (such as layer scaling during training) help ensure the positive impact of normalization on the robustness radius.

**Non-linearity of Feed-Forward Network Activations (GELU/ReLU/SwiGLU)**  The activation function determines the gain of the local Jacobian. In saturated regions (such as the left tail of GELU), the local slope approaches zero, which suppresses noise propagation and limits the norm of intermediate representations, thereby increasing the tolerable perturbation radius. In contrast, high-gain regions (resulting from large weights or strong inputs) amplify the norm of intermediate states and the output sensitivity, which translates to larger effective values for $\gamma$ and $C$. Gated variants (such as SwiGLU/MoE) can also trigger discrete switching of channels or experts near their thresholds, causing the output to undergo abrupt transitions in response to small perturbations. Overall, operating the activations in low-to-medium gain regions and controlling the scale of the weights helps to reduce the effective sensitivity and decrease $R_h$, which aligns with the monotonic properties described in Theorem 2.

**Residual Paths and Layer Scaling (Semantics of $\eta$)**    The residual path directly injects the representation from the previous layer into the next, scaled by a coefficient, which can be viewed as the $\eta$ in Theorem 2. A larger $\eta$ allows more input and intermediate perturbations to pass through without attenuation and accumulate in deeper layers, leading to an increase in the effective $\gamma$ and a decrease in the tolerable perturbation radius. Conversely, a smaller residual coefficient or layer scaling techniques (such as the ideas behind ReZero/LayerScale) can suppress this long-chain amplification and enhance robustness. A trade-off exists, as an overly small $\eta$ can limit feature reuse and gradient flow. In practice, a moderate but non-zero layer scaling is often adopted to achieve a better compromise between expressive power and robustness under the constraints of Theorem 2.

## D.2    EFFICIENCY OF OUR METHOD

In this section, we discuss the computational efficiency of the improved method we propose in §4.5. Let $M$ be the total number of candidate prompts, and let $T(\mathcal{M}, D)$ denote the time it takes for the model $\mathcal{M}$ to run once on the evaluation dataset $D$. Then, because calculating each candidate requires the hidden state vector for every data point, which necessitates a full inference pass, the total running time is:

$$O(M \cdot T(\mathcal{M}, D)) \tag{10}$$

Compared to other prompt optimization methods, many approaches also need to run each generated prompt on an evaluation dataset to assess its quality (e.g., TextGrad, OPRO). Therefore, the efficiency of our method is considered comparable to that of previous work. Furthermore, since this paper primarily focuses on theoretical analysis rather than methodological improvements, we leave further enhancements to the effectiveness and efficiency as future work.

## D.3    INPUT ROBUSTNESS OF NON-LINEAR SELF-ATTENTION

According to the discussion in §2.3, the certified input-perturbation radius obtainable via CoT depends on the model's Lipschitz properties. In this section, we replace the LSA with a *non-linear* (softmax) attention and derive a counterpart of Theorem 2.

We adopt a standard single-head attention mechanism with a residual flow. Let $W^Q, W^K, W^V, W^{PV} \in \mathbb{R}^{b \times b}$ be the projection matrices, and let $\tau > 0$ be the temperature. Denote the parameters by $\theta = (W^Q, W^K, W^V, W^{PV}, \tau)$ and let $E = [h, x]$. Define

$$Q = E\,W^Q, \qquad K = E\,W^K, \qquad V = E\,W^V, \qquad A(E) = \mathrm{softmax}\!\left(\tfrac{1}{\tau} Q K^\top\right),$$

where $\mathrm{softmax}$ is applied row-wise. To mitigate gradient explosion, we introduce a residual coefficient $\eta \in (0, 1)$ as in the LSA case. The non-linear attention map is

$$f_{\mathrm{Attn}}(h, x; \theta) \;=\; \eta\,E \;+\; W^{PV}\big(A(E)\,V\big). \tag{11}$$

In what follows, we analyze the input robustness of equation 11 under the same Lipschitz framework as in §2.3.

We first upper bound the two Lipschitz constants in equation 1. Throughout, $\|\cdot\|$ denotes the operator (spectral) norm.

**Lemma 2.** *Suppose $\|x\| \leq R_x$ and $\|h\| \leq R_h$. Let*

$$s_Q = \|W^Q\|, \quad s_K = \|W^K\|, \quad s_V = \|W^V\|, \quad s_{PV} = \|W^{PV}\|, \quad s_{QK} = s_Q s_K.$$

*Let $L_\sigma(\tau)$ be the (row-wise) Lipschitz constant of the $\mathrm{softmax}$ map with temperature $\tau$ under the chosen norm. Then the constants $C$ and $\gamma$ in equation 1 admit the bounds*

$$C \;\leq\; \eta \;+\; s_{PV}\, s_V\, L_\sigma(\tau)\, s_{QK}\, R_h^2, \qquad \gamma \;\leq\; \sqrt{\eta^2 \;+\; 4\, R_x^2 \big(s_{PV}\, s_V\, L_\sigma(\tau)\, s_{QK}\big)^2 R_h^2}.$$

Plugging Lemma 2 into the general input-perturbation bound equation 3, we obtain the following certified radius for non-linear attention.

**Theorem 3** (Certified Input-Perturbation Radius of Softmax Attention). *If $\|x\| \leq R_x$ and $\|h\| \leq R_h$, define*

$$\tilde{\beta} \;=\; s_{PV}\, s_V\, L_\sigma(\tau)\, s_{QK}\, R_h^2, \qquad \tilde{\gamma} \;=\; \sqrt{\eta^2 \;+\; 4\, R_x^2 \big(s_{PV}\, s_V\, L_\sigma(\tau)\, s_{QK}\big)^2 R_h^2}.$$

With $A > 0$ such that $\|e_0\| \le A\|\delta\|$, the certified tolerable input-perturbation radius of the map equation 11 at CoT step $K \in \mathbb{N}^+$ is

$$\|\delta\| \;\le\; \frac{(1 - \tilde{\gamma})\, R}{\left(\eta + \tilde{\beta}\right) \;+\; \left(A(1 - \tilde{\gamma})(1 + \tilde{\beta})\right) \tilde{\gamma}^K}.$$

In particular, if $\tilde{\gamma} < 1$, as $K \to \infty$,

$$\|\delta\| \;\le\; \frac{(1 - \tilde{\gamma})\, R}{\eta + \tilde{\beta}}.$$

It can be seen that Theorem 3 has a similar format to Theorem 2, showing that they have the same conclusion regarding the CoT robustness. The above discussion shows the effectiveness of our conclusion under the non-linear scenario.

*Proof.* Write $f_{\text{Attn}}(E) = \eta E + \Phi(E)$ with $\Phi(E) = W^{PV}\big(\text{softmax}(\frac{1}{\tau} E W^Q (E W^K)^\top)\, (E W^V)\big)$. By composing Lipschitz bounds of the bilinear map $E \mapsto E W^Q (E W^K)^\top$, the row-wise softmax (with constant $L_\sigma(\tau)$), and the linear maps $W^V, W^{PV}$, we obtain the stated bounds on $C$ and $\gamma$. Substituting them into equation 3 yields Theorem 3. □

### D.4 THE IMPACT OF VECTOR NORMS ON THE CoT ROBUSTNESS

While very small weight norms can indeed destabilize optimization during training, their analysis specifically targets inference-time robustness to input perturbations, so there is no contradiction—practical models must balance norm size for both stable training and robust inference. We further argue that robustness is better captured by absolute perturbations, i.e., the raw change in the output, rather than relative perturbations that normalize by the input magnitude, because the model ultimately makes decisions based on the absolute output vector. For example, in a multiple-choice setting, a substantial absolute shift in logits can change the predicted option even if the relative change is small. Therefore, we frame their conclusions in terms of absolute perturbation as a more faithful indicator of decision instability.

### D.5 ANALYSIS WITH MULTIPLE TOKENS

In our analysis, we treat the entire user query as a single embedding vector $x \in \mathbb{R}^d$, and the perturbation $\delta \in \mathbb{R}^d$ acts on this query-level representation rather than on individual token embeddings. The bivariate map $f(h, x)$ therefore models the interaction between (i) the current hidden state $h$ and (ii) a fixed embedding of the full input query, not between two literal tokens.

We now show that this is mathematically equivalent to starting from the usual multi-token transformer input.

**Corollary 1** (Equivalence of sequence input and bivariate model). *Let the model at a given CoT step take as input*

- *a hidden state $h \in \mathbb{R}^{d_h}$, summarizing all past reasoning tokens; and*

- *a sequence of question tokens with embeddings $(e_1, \dots, e_T)$, each $e_t \in \mathbb{R}^{d_e}$.*

*Assume its next-step hidden state is given by some deterministic map*

$$F_{\text{full}} : \mathbb{R}^{d_h} \times (\mathbb{R}^{d_e})^T \to \mathbb{R}^{d_h}.$$

*Then there exist*

1. *a linear isomorphism $U : (\mathbb{R}^{d_e})^T \to \mathbb{R}^d$ (vectorization / padding), and*

2. *a bivariate function $f : \mathbb{R}^{d_h} \times \mathbb{R}^d \to \mathbb{R}^{d_h}$,*

*such that the dynamics can be written exactly as*

$$h_{k+1} \;=\; f(h_k, x), \qquad x = U(e_1, \dots, e_T).$$

*Moreover, if $F_{full}$ is Lipschitz in $(h, (e_t)_t)$, then $f$ satisfies a Lipschitz condition of the form*

$$\|f(h_1, x_1) - f(h_2, x_2)\| \ \leq \ \gamma\|h_1 - h_2\| + C\|x_1 - x_2\| \tag{12}$$

*for some constants $\gamma, C \geq 0$.*

*Proof.* **Step 1: Reparameterizing the input sequence as a single vector.** Fix a maximum sequence length $T_{\max} \geq T$ and pad $(e_1, \ldots, e_T)$ with a distinguished padding embedding so that every input can be regarded as an element of $(\mathbb{R}^{d_e})^{T_{\max}}$. This space is linearly isomorphic to $\mathbb{R}^d$ with $d = d_e T_{\max}$.

Let

$$U : (\mathbb{R}^{d_e})^{T_{\max}} \to \mathbb{R}^d$$

be any fixed linear bijection (e.g., concatenation followed by a permutation of coordinates). Define

$$x = U(e_1, \ldots, e_T, \text{pad}, \ldots, \text{pad}) \in \mathbb{R}^d.$$

Conversely, $U^{-1}$ recovers the full token-level embedding tuple from $x$.

**Step 2: Defining the bivariate function.** Define

$$f(h, x) \ := \ F_{\text{full}}\big(h, U^{-1}(x)\big).$$

By construction,

$$h_{k+1} = F_{\text{full}}\big(h_k, (e_1, \ldots, e_T)\big) = f\big(h_k, U(e_1, \ldots, e_T)\big) = f(h_k, x),$$

so the original multi-token dynamics can be written as a bivariate map in $(h, x)$.

**Step 3: Preservation of Lipschitz continuity.** Suppose the model is Lipschitz in $(h, (e_t)_t)$, i.e., there exist constants $\gamma \geq 0$, $C_{\text{tok}} \geq 0$ such that for all $h_1, h_2$ and token sequences $(e_t), (e'_t)$,

$$\big\|F_{\text{full}}(h_1, (e_t)) - F_{\text{full}}(h_2, (e'_t))\big\| \ \leq \ \gamma\|h_1 - h_2\| + C_{\text{tok}} \big\|(e_t) - (e'_t)\big\|_{\text{seq}},$$

where $\|\cdot\|_{\text{seq}}$ is any norm on $(\mathbb{R}^{d_e})^{T_{\max}}$.

Using the linear isomorphism $U$, equip $\mathbb{R}^d$ with the induced norm

$$\|x\| := \big\|U^{-1}(x)\big\|_{\text{seq}}.$$

Then for any $(h_1, x_1), (h_2, x_2)$,

$$\begin{aligned}
\|f(h_1, x_1) - f(h_2, x_2)\| &= \big\|F_{\text{full}}(h_1, U^{-1}x_1) - F_{\text{full}}(h_2, U^{-1}x_2)\big\| \\
&\leq \gamma\|h_1 - h_2\| + C_{\text{tok}} \big\|U^{-1}(x_1) - U^{-1}(x_2)\big\|_{\text{seq}} \\
&= \gamma\|h_1 - h_2\| + C_{\text{tok}} \|x_1 - x_2\|.
\end{aligned}$$

Thus $f$ satisfies the Lipschitz condition with constants $\gamma$ and $C = C_{\text{tok}}$. $\qquad\square$

# E  ADDITIONAL INFORMATION

## E.1  EXPERIMENTAL DATASET

**MATH (Hendrycks et al., 2021)**  is a benchmark for competition-level mathematical reasoning, comprising $12,500$ problems with full step-by-step solutions ($7,500$ training and $5,000$ test). It spans diverse subfields (e.g., algebra, geometry, number theory, combinatorics, probability, and calculus) and is widely used to evaluate and distill chain-of-thought style reasoning in mathematics. In this paper, we evaluate our conclusions with the subset of MATH, which contains $500$ data following Lightman et al. (2024).

**MMLU-Pro (Wang et al., 2024c)**  It is a strengthened successor to MMLU that emphasizes higher question quality and robustness. It contains over $12,000$ multiple-choice questions drawn from textbooks and exams across $14$ academic domains (e.g., biology, business, chemistry, computer science, economics, engineering, health, history, law, mathematics, philosophy, physics, psychology, and others). Each item offers $10$ options, which reduces guessability and increases discrimination among strong models.

**GPQA (Rein et al., 2024)**  targets graduate-level, "Google-proof" scientific reasoning. The test set includes $448$ expert-authored multiple-choice questions in biology, physics, and chemistry, designed such that even with open-web access, non-experts struggle while domain experts achieve only modest accuracy. GPQA thus probes high-level knowledge, multistep reasoning, and model reliability under stringent oversight conditions.

## E.2   PROMPT

---
**Prompt of MATH**

---
Solve the following math problem efficiently and clearly.
Regardless of the approach, always conclude with:
Therefore, the final answer is: $boxed{answer}$. I hope it is correct.

---
**Prompt of MMLU-Pro**

---
The following are multiple choice questions (with answers) about domain.
Think step by step and then finish your answer with "the answer is (X)" where X is the correct letter choice.

---
**Prompt of GPQA**

---
Given the following question and four candidate answers (A, B, C and D), choose the best answer.

- For simple problems:
Directly provide the answer with minimal explanation.

- For complex problems:
Use this step-by-step format:
## Step 1: [Concise description]
(Brief explanation)
## Step 2: [Concise description]
(Brief explanation)

Regardless of the approach, always conclude with:
The best answer is [the_answer_letter].
where the [the_answer_letter] is one of A, B, C or D.

Let's think step by step.

---

Table 4: The prompt used in this paper.

In this section, we list the prompt we used in Table 4.

## E.3   PROMPT NUMBER OF EACH SETTING

In this section, we present the number of prompts used for each dataset and model, as shown in Table 5. From the table, we can observe that the number of prompts is not consistent across different settings. This is because, during prompt optimization, the suitable prompts vary for different models and datasets. To ensure that optimal performance is achieved for each setting, we use a different set of prompts for each setting.

| Dataset | Llama2-7b | Llama3.1-8b | Llama-R1-8b | Qwen3-8b |
|---------|-----------|-------------|-------------|----------|
| MATH    | 14        | 29          | 18          | 13       |
| MMLU-Pro | 20       | 29          | 20          | 16       |
| GPQA    | 11        | 20          | 16          | 12       |

Table 5: The total number of generated prompts using TextGrad, OPRO, and CFPO under each setting.

### E.4 CALCULATION OF OUTPUT FLUCTUATION

Consider a collection of model outputs produced for the same input, represented as a multiset of strings of size $M$. Let $p_i$ denote the empirical frequency of the $i$-th distinct string. The metric computes the Shannon entropy:

$$H = -\sum_i p_i \log_2 p_i, \tag{13}$$

and normalizes it by the maximal entropy achievable with $M$ samples, namely $\log_2 M$. The resulting index:

$$\widehat{H} = \frac{H}{\log_2 M} \in [0, 1] \tag{14}$$

is scale-free and directly comparable across different sample sizes. By construction, $\widehat{H} = 0$ when all outputs are identical (complete consensus, no fluctuation) and $\widehat{H} = 1$ when all $M$ outputs are distinct (maximal dispersion, each outcome occurs once). For empty or singleton sets, the metric is defined to be 0, reflecting the absence of observable variability.

Output fluctuation manifests as dispersion in the empirical outcome distribution. Greater variability spreads probability mass more evenly across distinct strings, driving $H$ toward its maximum and increasing $\widehat{H}$. Greater stability concentrates mass on a single outcome, driving $H$ toward 0 and decreasing $\widehat{H}$. Normalization by $\log_2 M$ ensures that the same qualitative level of dispersion yields comparable scores even when the number of samples differs, while preserving the desired extremes ("all same" $\rightarrow 0$, "all different" $\rightarrow 1$).

### E.5 IMPLEMENTATION DETAILS

The input and hidden state vectors used in our experiments are the encoded vectors from the embedding layer and the final layer of the respective LLMs for the corresponding inputs. For each input, we set the model to generate a single output, with the temperature set to 0, top_p to 1.0, and the random seed fixed at 42. Our experiments are run on a single A100-80G GPU, with the average experiment time for each setting being approximately one hour. All our codes are implemented with PyTorch (Paszke et al., 2019), Transformers (Wolf et al., 2020), and VLLM (Kwon et al., 2023) using Python3.10. We detail how to plot the analysis figure in Appendix E.6.

### E.6 PLOT OF ANALYSIS FIGURE

We visualize conditional distributions with an $x$–binned box–plot design. The $x$–range is uniformly partitioned into $K = 10$ equal–width bins; within each bin we compute the first quartile ($Q_1$), median, and third quartile ($Q_3$). Whiskers follow Tukey's rule and extend to the most extreme observations within $[\, Q_1 - 1.5\,\mathrm{IQR},\ Q_3 + 1.5\,\mathrm{IQR}\,]$, where $\mathrm{IQR} = Q_3 - Q_1$; bins with very few points are shown by a median marker only.

To convey the trend across bins, the binwise medians are connected by a shape–preserving piecewise cubic Hermite interpolant (PCHIP). An optional interquartile ribbon is drawn by interpolating $Q_1$ and $Q_3$ with the same scheme. For context, we overlay lightly jittered raw points in the background and add marginal density curves along the top (for $x$) and the right (for $y$), estimated via Gaussian KDE with Silverman's bandwidth; the right–hand marginal can be computed from an alternative $y$ sample when provided. Box widths adapt to local bin spacing to prevent overlap in narrow $x$–ranges, and a unified low–saturation color palette is used for visual consistency.

## F ADDITIONAL EXPERIMENT

### F.1 FITTING $\gamma$ OF THEOREM 1

In this section, we verify that $\gamma < 1$ to ensure the reliability of the conclusions derived from Equation 4. Since the right-hand side of the inequality in Theorem 1 is positively correlated with $\gamma$, we consider the extreme case by replacing the inequality with an equality, which gives:

$$\|\varepsilon_K\| = \left( A\gamma^K + \frac{C}{1-\gamma}(1-\gamma^K) \right) \|\delta\| \tag{15}$$

| Model | MATH | MMLU-Pro | GPQA |
|-------|------|----------|------|
| Llama2-7b | 0.662 | 0.892 | 0.671 |
| Llama3.1-8b | 0.476 | 0.218 | 0.014 |
| Llama-R1-8b | 0.879 | 0.896 | 0.871 |
| Qwen3-8b | 0.754 | 0.744 | 0.015 |

Table 6: The fitted $\gamma$ on different models and datasets.

| Model | Amazon | | FinQA | | ToolE | |
|-------|--------|----|-------|----|-------|----|
| | EM | OF | EM | OF | EM | OF |
| Llama2-7b | $17.4 \pm 11.9$ | 0.711 | $5.9 \pm 3.7$ | 0.479 | $27.1 \pm 17.6$ | 0.383 |
| Llama3.1-8b | $61.1 \pm 35.6$ | 0.271 | $37.6 \pm 6.1$ | 0.377 | $49.8 \pm 18.0$ | 0.365 |
| Llama-R1-8b | $60.3 \pm 39.8$ | 0.242 | $45.7 \pm 7.4$ | 0.276 | $51.5 \pm 4.9$ | 0.162 |
| Qwen3-8b | $61.1 \pm 18.4$ | 0.201 | $54.9 \pm 5.9$ | 0.246 | $56.0 \pm 6.6$ | 0.121 |

Table 7: The performance on Amazon Rview (Amazon), FinQA, and ToolE.

Then, for each question across all datasets, we compute the corresponding $\|\delta\|$ and $\|\varepsilon_K\|$ for different CoT steps $K$ among all generated answers. We use this data to fit the parameter $\gamma$ in Equation 15 using the least squares method. The fitting results are shown in Table 6. From the table, we can observe that the value of $\gamma$ is less than 1 in all settings, which validates the reliability of the assumption made in our analysis.

## F.2 Performance on More Datasets

To more comprehensively validate the changes in output fluctuation across different datasets, we conduct experiments on a broader range of datasets. We conduct experiments on the Amazon Review (Ni et al., 2019) (sentiment analysis), FinQA (Chen et al., 2021) (financial question answering), and ToolE (Huang et al., 2024) (tool use) datasets to verify our conclusions in scenarios that more closely resemble real-world applications. The experimental results are presented in Table 7. From the table, we can observe that as the model performance improves, the output fluctuation shows an overall downward trend. This is consistent with the conclusions we draw in Table 2.

## F.3 Output Fluctuation with Other Metric

To more comprehensively evaluate fluctuations in model outputs, this section quantifies semantic variability (SV) across models and datasets. For each question, we first compute an embedding vector for every answer using all-MiniLM-L6-v2 (Wang et al., 2020b). We then take the average distance from these vectors to their centroid (mean vector) as the metric of output variability. As shown in Table 8, this metric exhibits strong agreement with OF, and the experimental findings are consistent with those in Table 8, thereby corroborating the correctness of our theoretical analysis.

## F.4 Performance with Same Prompts

To ablate the effect of prompt differences on the evaluation, we conduct experiments using the same prompts across all models and datasets. For all models, we employ the prompts generated with Qwen3-8b. The results, as shown in Table 9, indicate that the conclusions drawn from using

| Model | MATH | | MMLU-Pro | | GPQA | |
|-------|------|----|----------|----|------|----|
| | SV | OF | SV | OF | SV | OF |
| Llama2-7b | 0.851 | 0.475 | 0.706 | 0.622 | 0.760 | 0.509 |
| Llama3.1-8b | 0.777 | 0.366 | 0.631 | 0.350 | 0.667 | 0.467 |
| Llama-R1-8b | 0.669 | 0.158 | 0.601 | 0.292 | 0.575 | 0.371 |
| Qwen3-8b | 0.653 | 0.097 | 0.579 | 0.162 | 0.554 | 0.214 |

Table 8: The output fluctuation using different metrics.

| Model | MATH | | MMLU-Pro | | GPQA | |
|---|---|---|---|---|---|---|
| | EM | OF | EM | OF | EM | OF |
| Llama2-7b | 12.2 | 0.653 | 13.8 | 0.578 | 15.9 | 0.523 |
| Llama3.1-8b | 48.9 | 0.375 | 35.1 | 0.347 | 28.1 | 0.470 |
| Llama-R1-8b | 65.4 | 0.147 | 40.2 | 0.303 | 30.0 | 0.404 |
| Qwen3-8b | 77.2 | 0.097 | 46.9 | 0.162 | 37.3 | 0.214 |

Table 9: The average EM and OF of different models and datasets. For the certain dataset, the prompts of each model are all same with Qwen3-8b.

| Model | Scale | MATH | | MMLU-Pro | | GPQA | |
|---|---|---|---|---|---|---|---|
| | | EM | OF | EM | OF | EM | OF |
| Llama3.1 | 8b | $45.8 \pm 7.2$ | 0.366 | $41.0 \pm 10.7$ | 0.350 | $26.6 \pm 5.7$ | 0.467 |
| | 70b | $56.0 \pm 12.8$ | 0.284 | $63.0 \pm 14.4$ | 0.186 | $42.6 \pm 9.7$ | 0.232 |
| Qwen3 | 8b | $77.2 \pm 1.6$ | 0.097 | $46.9 \pm 5.2$ | 0.162 | $37.3 \pm 1.9$ | 0.214 |
| | 34b | $80.8 \pm 4.0$ | 0.075 | $67.8 \pm 5.1$ | 0.104 | $43.6 \pm 3.2$ | 0.177 |

Table 10: The performance of Llama3.1 and Qwen3 on each dataset with different model scales.

identical prompts are consistent with those in Table 2. Therefore, in our main experiments, to ensure a consistent methodology, we use each model to generate the prompts for its own inference.

### F.5 PERFORMANCE CROSS DIFFERENT MODEL SCALE

To verify how output fluctuations change with input perturbations on models of different scales, we measure the performance of models of varying scales on each dataset. The experimental results are shown in Table 10. From the table, we can find that although the EM of larger-scale models could exhibit greater fluctuations, from the perspective of OF, larger-scale models generally demonstrate better input robustness. This is because larger-scale models tend to generate a greater number of reasoning steps $K$ (Wei et al., 2022; Kojima et al., 2022) and possess a stronger ability to widen the confidence gap between correct and incorrect answers, which in turn increases the acceptable perturbation threshold $R$ (Zhu et al., 2023; Chhikara, 2025). Consequently, according to Theorem 3, larger-scale models exhibit better robustness.

### F.6 PERFORMANCE WITH CoT STEPS UNDER EACH SETTING

In this section, we list how performance varies with CoT steps under different models of MATH in Figure 6.

### F.7 PERTURBATION WITH EMBEDDING NORM UNDER DIFFERENT SETTINGS

The variation of output perturbation with respect to embedding norm for all models on various datasets is illustrated in Figure 7 to Figure 9.

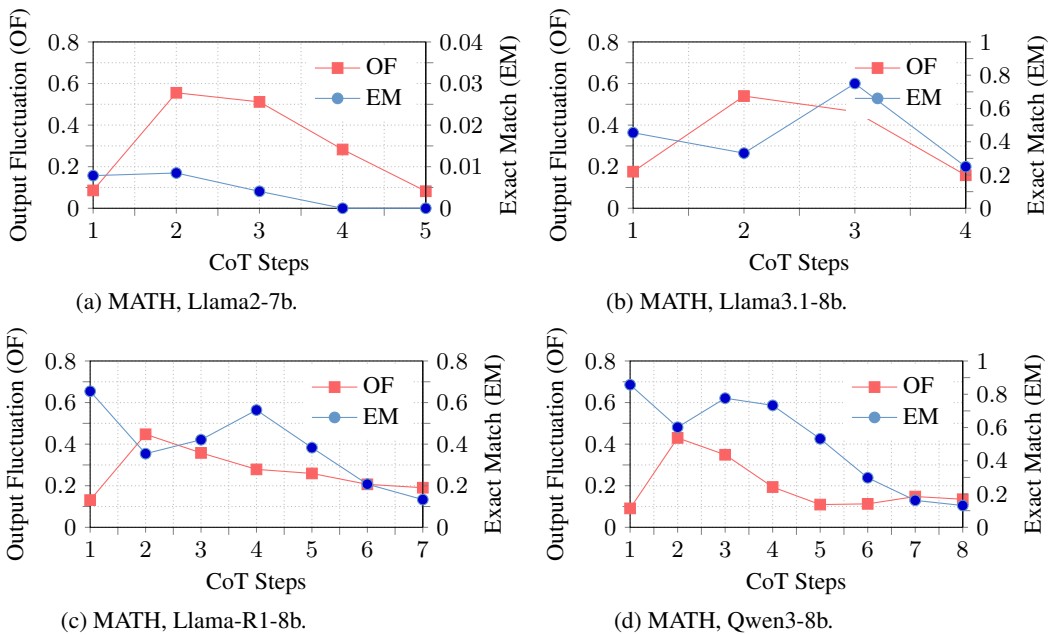

Figure 6: EM and OF on MATH cross CoT steps with different models.

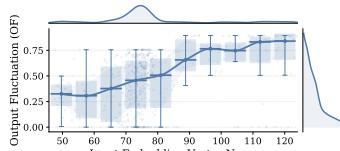

Figure 7: The change in output fluctuation with the norm of the input embedding vector across all experimental models on MATH. Each point denotes the result of one question, where X-axis denotes the input vector norm and Y-axis denotes OF of this question. The Pearson coefficient is $0.415$.

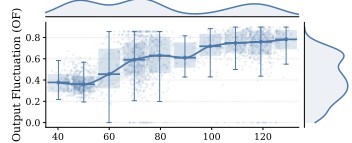

Figure 8: The change in output fluctuation with the norm of the input embedding vector across all experimental models on MMLU-Pro. Each point denotes the result of one question, where X-axis denotes the input vector norm and Y-axis denotes OF of this question. The Pearson coefficient is $0.634$.

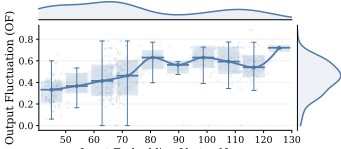

Figure 9: The change in output fluctuation with the norm of the input embedding vector across all experimental models on GPQA. Each point denotes the result of one question, where X-axis denotes the input vector norm and Y-axis denotes OF of this question. The Pearson coefficient is $0.541$.

