# OpenReview forum: "Bounds of Chain-of-Thought Robustness: Reasoning Steps, Embed Norms, and Beyond"
_ICLR.cc/2026/Conference — ICLR 2026 Poster_

### Official Review · Reviewer_Kkzt · 2025-10-21

**Soundness:** 2
**Presentation:** 3
**Contribution:** 3
**Rating:** 4
**Confidence:** 4

**Summary:**

This paper provides a theoretical investigation into the robustness of Chain-of-Thought (CoT) against input perturbations, supported by empirical tests on a range of open-source models to validate their arguments. The authors model the CoT process as a bivariate Lipschitz function and leverage classical stability results of Lipschitz functions to derive stability relationships for the Linear Self-Attention (LSA) model. Through theoretical analysis and experimental validation, the authors find that: (1) model performance is positively correlated with robustness; (2) longer reasoning chains reduce the impact of input perturbations, but cannot eliminate it entirely; and (3) the norms of the embedding and hidden state vectors are negatively correlated with model robustness. The paper also proposes a novel prompt optimization method aimed at enhancing model robustness.

**Strengths:**

1. This paper rigorously derives bounds on output fluctuation under input perturbation, building from a Lipschitz continuity framework (Eq. 1, Thm. 1) to a specialized analysis of Linear Self-Attention (LSA). This progression yields novel insights, crucially linking perturbation robustness to the norms of input and hidden state embeddings (Thm. 2).
2. The experimental evaluation is comprehensive, conducted across multiple datasets and state-of-the-art LLMs. The results, notably in Table 1 and Figures 1-3, provide compelling quantitative evidence for the paper's central claims by directly visualizing the relationships between input perturbation, reasoning steps, and output fluctuation.

**Weaknesses:**

1. The paper models the LLM's reasoning process directly as a bivariate function. However, in reality, the input to an LLM during reasoning clearly consists of more than two tokens. The authors need to more clearly demonstrate why this model holds.
2. The authors primarily provide a theoretical analysis for the Linear Self-Attention (LSA) model in the main text, while the discussion of other modules is relegated to the appendix and remains rather superficial. A more in-depth discussion on the impact of these other components on the theoretical results is necessary, or alternatively, the authors should justify why the LSA findings are sufficient for analyzing the entire model.
3. In the experimental section, the authors need to clearly describe their procedure for perturbing the input prompts and explain how they modify the norms of the embedding and hidden state vectors.
4. In the results shown in Figures 2 and 3, the model performance declines as the length of the Chain-of-Thought increases. This suggests that, under the experimental setup of this paper, extending the number of CoT steps may no longer be beneficial. Consequently, the validity of the conclusions related to Output Fluctuation is called into concern. The authors should identify more suitable tasks where increasing CoT steps demonstrably improves model performance, and then examine the behavior of OF under such conditions.
5. The results presented in Figures 4 and 5 do not provide compelling support for the authors' claims. Particularly in Figure 4, aside from the notable jump between 60-70, the remaining data points lack a clear monotonic increasing trend.
6. The paper identifies that larger embedding and hidden state norms reduce CoT robustness, a finding that appears counterintuitive from a practical training perspective. Standard techniques that effectively constrain these norms (e.g., aggressive weight decay, smaller initialization) are well-known to potentially harm training stability. Furthermore, if the absolute magnitude of the perturbation is held constant, a smaller vector norm would intuitively represent a larger *relative* perturbation, which one might expect to be more detrimental to robustness. The authors should provide a more intuitive explanation and discussion to reconcile this theoretical finding with practical intuition. For instance, is the analysis sensitive to the assumption that the perturbation is absolute rather than relative? A discussion on this apparent discrepancy is crucial for the audience to properly interpret and trust the theoretical results.

**Questions:**

See weakness.

---

> ### Author Response · Authors · 2025-11-15
> **Official Comment by Authors (1/2)**
>
> Thank you for your positive assessment of our work! **We have revised the paper according to your review; all changes are highlighted in blue**. Below, we respond to your specific comments:
>
> 1. *The input to an LLM during reasoning clearly consists of more than two tokens. (**Lines 99-101 and Appendix D.5 in Revision**)*
> - **We have proved in the revision that even if the input consists of multiple tokens, the conclusion is consistent with the current paper, as discussed in Appendix D.5**.
> - Besides, this paper models the LLM’s reasoning update as a function $f(h,x)$ of two vector inputs. In our setting, the inputs $x$ and $h$ are not single-token embeddings but rather the full semantic vector of the user query and the hidden-state vector, respectively. Since our analysis focuses on how the entire query influences the iterative reasoning dynamics rather than token-level interactions, this abstraction follows a common simplification in the literature [1-3]. Besides, [3] has proven that the trained Transformer first encodes the input tokens into an internal representation that facilitates gradient descent (or its preconditioned version) within the context. Therefore, we suppose that it is appropriate for our purposes.
>
> 2. *The discussion of other modules of the Transformer is relegated to the appendix and remains rather superficial. (**Appendix D.3 in Revision**)*
> - The revised paper now **includes a discussion of nonlinear settings in Appendix D.3**: nonlinearity primarily affects the upper bound of the Lipschitz constant in Lemma 1 but does not change our final conclusions. Empirically, Section 4 shows that in nonlinear attention models (e.g., Llama and Qwen), the results align with our theory, further validating the correctness in nonlinear scenarios.
> - Besides, prior work [4,5] has shown that the conclusions drawn from simplified Transformer variants can reflect the properties of full Transformers. Consistent with this, our Section 4 experiments demonstrate that the results on real LLMs match our theoretical predictions, supporting the reliability of LSA-based conclusions for full Transformers.
>
> 3. *Authors need to clearly describe their procedure for perturbing the input prompts and explain how they modify the norms of the embedding and hidden state vectors. (**Lines 304-311**)*
> - **As discussed in Lines 304-311**, we perturb inputs by rewriting the prompts. Specifically, we adopt established prompt-optimization methods (TextGrad, OPRO, and CFPO) to ensure the quality of the rewritten prompts. Because the prompts change, the corresponding input embeddings and hidden states also change, leading to different vector norms.
>
> 4. *In the results shown in Figures 2 and 3, the model performance declines as the length of the Chain-of-Thought increases. Consequently, the validity of the conclusions related to Output Fluctuation is called into question. (**Appendix F.6 in Revision**)*
> - To more finely examine how EM and OF change with the number of CoT steps, **we report the results of various models on MATH in Appendix F.6**. We observe that under different settings, as the number of CoT steps increases, the overall performance first rises and then falls (except for the simple problems corresponding to Step = 1), which is consistent with our observations in Figure 2 and 3. The initial increase indicates that CoTs does indeed help enhance reasoning performance, while the later decrease occurs because very long CoTs are generated only for particularly difficult problems. Thus, so the performance on problems associated with long CoTs is generally weaker than on those associated with short CoTs. Moreover, **prior work has also shown that as the CoT step increases, the performance of LLMs degrades [1,2].** Therefore, we believe there is hardly any mainstream dataset on which increasing the number of CoT steps can significantly improve performance.
> - Meanwhile, as the number of CoT steps increases, OF shows a consistently decreasing trend. We believe this is because model **performance and robustness are two different dimensions**. In Figure 2 and Figure 3, even for problems corresponding to long CoT steps where the model cannot produce the correct answer, the OF between the incorrect answers generated by different prompts is low, i.e., they are “**consistently wrong**.” We take this as evidence supporting our conclusion that longer CoT steps reduce output fluctuation.

---

> > ### Author Response · Authors · 2025-11-15
> > **Official Comment by Authors (2/2)**
> >
> > 5. *The results presented in Figures 4 and 5 lack a clear monotonic increasing trend.*
> > - As noted in the captions of Figures 4 and 5, the **Pearson correlation coefficients are $0.506$ and $0.229$**, respectively, indicating an overall increasing trend. The trend may appear weak, largely because the y-axis spans a broad range. More concretely, **the means in both figures rise from about $0.25$ to $0.50$, leading to a $\sim 100\%$ increase**, which shows a clear overall upward trend despite local non-monotonicities.
> >
> > 6. *A discussion of this apparent discrepancy between the theoretical finding and practical intuition is crucial for the audience to properly interpret and trust the theoretical results. (**Appendix D.4 in Revision**)*
> > - In the revision, **we have updated the discussion of the vector norm impact in Appendix D.4**, including:
> >     - **Small norms causing training instability**. We suppose that **training stability and CoT robustness are two different things**. Thus, there is no contradiction: in practice, one must balance norm magnitudes to ensure both training stability and inference robustness.
> >     - **Absolute vs. relative perturbations**. We argue that absolute perturbation (measuring the magnitude of the output change itself) is a more faithful measure of robustness than relative perturbation (output change normalized by input magnitude), because **the model’s decision is made on the absolute output vector, not on “output divide input”**. For instance, in multiple-choice settings, a large absolute shift in the output logits can flip the selected option even if the input vector is large, which corresponds to a large absolute but small relative perturbation. Hence, absolute perturbation better reflects decision instability, and our conclusions are framed in these terms.
> >
> > [1] Trained Transformer Learn Linear Models In-Context. Zhang, et al., PMLR.
> >
> > [2] Asymptotic theory of ICL by linear attention. Lu, et al., ArXiv.
> >
> > [3] Transformers learn in-context by gradient descent. Oswald, et al., ICML 2023.
> >
> > [4] Trained Transformer Learn Linear Models In-Context. Zhang, et al., PMLR.
> >
> > [5] Linformer: Self-Attention with Linear Complexity. Wang, et al., ArXiv.
> >
> > [6] When More is Less: Understanding Chain-of-Thought Length in LLMs. Wu, et al., 2025.
> >
> > [7] Towards Reasoning Era: A Survey of Long Chain-of-Thought for Reasoning Large Language Models. Chen et al., 2025.

---

> > > ### Comment · Reviewer_Kkzt · 2025-11-26
> > >
> > > Thanks for the author's response. Given that the author has made revisions to the article addressing the issues I raised, I will increase my score. However, I still believe that Figure 4 does not adequately support the conclusions drawn in Section 4.4. Perhaps the author could attempt to perturb the norm of the embedding vector over a larger range or move this part of the discussion to the appendix.

---

> > > > ### Author Response · Authors · 2025-11-26
> > > >
> > > > Thank you for your suggestion! Considering that increasing the norm of the embedding vector would lead to inconsistencies in the experimental setup for the analysis, in the new revision we instead report in **Appendix F.7** the change in output fluctuation with respect to the norm of the input embedding vector for each dataset. The Pearson correlation coefficients for MATH, MMLU-Pro, and GPQA are 0.415, 0.634, and 0.541, respectively, showing the expected positive correlation. Moreover, there is also a clearly observable increasing trend from a qualitative perspective.

---

### Official Review · Reviewer_4uQH · 2025-10-25

**Soundness:** 3
**Presentation:** 2
**Contribution:** 2
**Rating:** 4
**Confidence:** 4

**Summary:**

The authors propose a theoretical analysis of how input perturbations propagate through CoT reasoning in LLMs. It derives upper bounds on output fluctuations under Lipschitz continuity, showing that: (i) increasing reasoning steps mitigate, but never remove, the impact of input perturbations; and (ii) robustness is negatively correlated with the norms of input embeddings and hidden state vectors. The analysis is instantiated on a Linear Self-Attention model, treated as a simplified Transformer, and validated through experiments on four models and three benchmarks. Empirical results align with theory, confirming that longer reasoning chains dampen but do not eliminate perturbation sensitivity. The authors also propose a simple prompt-pick strategy that maximises input-perturbation tolerance, yielding modest improvements.

**Strengths:**

- Theoretical grounding: The paper offers one of the first formal derivations of CoT robustness bounds, establishing links between perturbations, reasoning steps, and embedding norms.


- Clear empirical validation: Experiments on diverse datasets and models convincingly support theoretical claims.

- Insightful analysis of LSA: Using Linear Self-Attention provides tractable yet informative insights into Transformer dynamics.

**Weaknesses:**

- Simplified modelling assumptions: The reliance on LSA as a proxy for real Transformers oversimplifies key non-linear dynamics (softmax attention, layer normalisation).

- Lack of deeper interpretability: While the derivations are sound, the paper does not explore the mechanistic implications for CoT reasoning or noise propagation in practical LLMs.

- Marginal methodological contribution: The proposed prompt optimisation method adds limited value and feels auxiliary to the theoretical core.

- Lack of transparency: The authors could have shared the code in order to improve reproducibility and be transparent

**Questions:**

- Theoretically, why should the negative correlation between robustness and embedding norms hold across non-linear, attention-based architectures?

- Beyond quantifying bounds, how might your framework inform practical interventions to improve CoT robustness during training or inference?

---

> ### Author Response · Authors · 2025-11-15
>
> Thank you for recognizing our work! **We have revised the paper in light of your review, with all changes highlighted in blue**. Below are our responses to your questions:
>
> 1. *Simplified modeling assumptions (**Appendix D.3 in Revision**)*
> - We adopt LSA in Section 3 because a substantial body of prior analysis uses this simplified setting to obtain transparent bounds [1-3] while preserving key behaviors of full Transformers. In our revision, we also extend the analysis beyond LSA: **Appendix D.3 replaces linear attention with standard softmax attention** and derives parallel Lipschitz-based bounds. In this nonlinear setting, the nonlinearity changes the Lipschitz constants but not the qualitative form of the robustness guaranties; the core conclusions, therefore, remain intact. Consistently, Section 4 reports results on non-linear attention models (e.g., Llama and Qwen) that align with our theoretical predictions.
>
> 2. *Lack of deeper interpretability (**Lines 131-136 in Revision**)*
> - Our focus is not on how LLMs implement CoT, but on how sensitive CoT outputs are to prompt perturbations. We present several factors affecting the robustness of CoT. Theorem 1 shows that output fluctuation arises from two terms: (i) an accumulated term through the hidden state that is iteratively scaled by the hidden state Lipschitz factor $\gamma$; and (ii) an input-retained term governed by the input Lipschitz constant $C$. This decomposition formalizes how perturbations propagate and accumulate across steps.
>
> 3. *Marginal methodological contribution (**Lines 455-458**)*
> - As discussed in Section 4.5, **our aim is not to introduce a new prompt optimizer per se, but to empirically validate the theoretical analysis**: we select prompts by maximizing the certified upper bound implied by our analysis, which yields consistent improvements over prior baselines. The methodological component is thus intentionally lightweight and primarily serves as a validation of the theoretical conclusions.
>
> 4. *Lack of transparency (**Line 53 in Revision**)*
> - We appreciate the request. **The revised manuscript includes an anonymized repository to facilitate reproducibility in Line 53 and we have updated the supplementary material**.
>
> 5. *Theoretically, why should the negative correlation between robustness and embedding norms hold across non-linear, attention-based architectures? (**Appendix D.3 in Revision**)*
> - **We add Appendix D.3 in the revision to establish non-linear analogs of the Lipschitz bounds for standard softmax attention**. In that setting, the constants $C$ and $\gamma$ depend on model/operator norms and the softmax Lipschitz constant, where the bound retains the same structure as Theorem 2. Consequently, the qualitative dependence of robustness on input and hidden-state norms carries over. Empirically, models of Section 4 (Llama, Qwen) corroborate this negative correlation.
>
> 6. *How might your framework inform practical interventions to improve CoT robustness during training or inference? (**Lines 261-265 in Revision**)*
> - **Inference time**. Theorem 1 implies that longer CoT reduce output fluctuation; thus, encouraging longer, structured chains can enhance robustness. In addition, optimize prompts to decrease the norms of the input embedding and of the encoded vectors lowers the effective Lipschitz factors entering the bound (cf. Lemma 1 and Theorem 2), thereby improving robustness. This is precisely the rationale behind our prompt-selection criterion in Section 4.5.
> - **Training time**. Theorem 2 indicates that reducing the norms of encoded representations and decreasing the training-data covariance $\Gamma$ (i.e., improving data consistency) increases the certified perturbation radius. Due to computational constraints, we leave full training-time interventions to future work, but the theory specifies the levers: shrink representation norms, reduce $\Gamma$ , and choose moderate layer scaling.
>
> [1] Trained Transformer Learn Linear Models In-Context. Zhang, et al., PMLR.
>
> [2] Linformer: Self-Attention with Linear Complexity. Wang, et al., ArXiv.
>
> [3] Training Dynamics of In-Context Learning in Linear Attention. Zhang, et al., ICML 2025.

---

> ### Author Response · Authors · 2025-11-28
>
> Dear Reviewer, we have revised the paper based on your review and submitted a response, and we sincerely hope that our changes address your concerns. As there are now fewer than five days remaining before the rebuttal period ends, we would be truly grateful for your feedback and sincerely look forward to your reply.

---

### Official Review · Reviewer_K3vW · 2025-10-28

**Soundness:** 2
**Presentation:** 2
**Contribution:** 2
**Rating:** 6
**Confidence:** 2

**Summary:**

This paper provides an analysis on the robustness of CoT reasoning w.r.t. input perturbations. The authors derive output upper bounds on how input perturbations propagate through CoT reasoning, claiming that increasing the number of reasoning steps reduces the fluctuation, but it can't be fully eliminated. Besides they show that robustness is negatively correlated with input and hidden state norms.

**Strengths:**

Providing a formal mathematical model of CoT perturbation propagation and finding an upper bound on robustness.

**Weaknesses:**

From what i understand the Lipschitz continuity assumption is an over simplifying assumption

**Questions:**

I am not sure about this but the authors considered multiple assumption to justify their modeling. Is there any prove that these assumption doesn't make the results to diverge from real cases?

---

> ### Author Response · Authors · 2025-11-15
>
> Thank you for your positive assessment of our work! **We have revised the paper in response to your review, and all changes are highlighted in blue**. Below, we address your specific questions point by point.
>
> 1. *From what I understand, the Lipschitz continuity assumption is an oversimplification. (**Lines 116-118**)*
> - As discussed in Lines 116-118, Lipschitz continuity is a standard and **widely adopted modeling assumption** in recent theoretical analyzes of LLMs [1-3], where many works have proven it [4,5]. Our analysis follows this common practice, which we believe is appropriate for the problem at hand.
>
> 2. *Is there any proof that these assumptions do not cause the results to diverge from real cases? (**Appendix D.3 and Appendix D.5 in Revision**)*
> - Beyond Lipschitz continuity, our work uses two additional assumptions and provides evidence that they are reasonable in practice:
>     - A contraction on intermediate states (Lipschitz constant $\gamma < 1$). We justify $\gamma < 1$ in line with prior stability analyzes and empirically fit $\gamma$ across settings (Appendix F.1), supporting the assumption from a practical standpoint.
>     - Linearization of the Transformer via LSA. LSA is a conventional simplification used to analyze Transformer behavior [6, 7]; we adopt it for clarity and tractability. We also discuss the influence of non-linear components and analyze a non-linear attention model, showing that our main conclusions continue to hold in the non-linear setting (Appendix D.3). Moreover, the empirical results in Section 4 agree with our theoretical predictions, reinforcing the validity of this modeling choice.
> - In summary, across four mainstream LLMs and multiple reasoning datasets, the experimental trends match our theoretical analysis, **indicating that our assumptions do not cause the results to diverge from real cases**.
>
> [1] In-context learning with transformers: softmax attention adapts to function lipschitzness. Collins, et al., NeurIPS 2024.
>
> [2] Rethinking Lipschitz Neural Networks and Certified Robustness. Zhang, et al., NeurIPS 2022.
>
> [3] Mitigating Transformer Overconfidence via Lipschitz Regularization. Ye, et al., PMLR.
>
> [4] LipsFormer: Introducing Lipschitz Continuity. Qi, et al., ArXiv.
>
> [5] A New Local Lipschitz Bound for Transformers. Yudin, et al., ArXiv.
>
> [6] Trained Transformer Learn Linear Models In-Context. Zhang, et al., PMLR.
>
> [7] Linformer: Self-Attention with Linear Complexity. Wang, et al., ArXiv.

---

> > ### Comment · Reviewer_K3vW · 2025-11-27
> >
> > Thank you for your clarifications. I will maintain my original score as my initial assessment was made regardless of the mentioned points.

---

> > > ### Author Response · Authors · 2025-11-27
> > >
> > > Thank you very much for your reply! If our rebuttal has partially addressed your concerns, could you please increase your **confidence**? Thank you again!

---

### Official Review · Reviewer_M3bE · 2025-10-30

**Soundness:** 3
**Presentation:** 2
**Contribution:** 3
**Rating:** 6
**Confidence:** 3

**Summary:**

This paper studies the robustness of CoT reasoning and theoretically proves that while increasing reasoning steps can partially mitigate the effect of input perturbations, even infinitely long chains cannot completely eliminate them. Further analysis on the simplified Linear Self-Attention model shows that larger input embedding and hidden state norms reduce robustness. Experiments on several mainstream large models and reasoning datasets validate the theory, and a new prompt selection method based on norm minimization is proposed to effectively enhance robustness.

**Strengths:**

This paper provides a formal theoretical derivation of the robustness of CoT reasoning and establishes an upper bound on output fluctuations under input perturbations.

**Weaknesses:**

1. This paper primarily adopts OF as the robustness metric, but does not account for semantic-level robustness. Could this limitation lead to an underestimation of the model’s actual robustness?

2. This paper does not specify the exact prompt templates used in the experiments. Could the authors clarify the prompt design choices?

3. The analysis suggests that larger input embedding norms reduce robustness. However, given that different models vary in embedding initialization and pretraining corpora, how generalizable is this finding across architectures?

**Questions:**

see weaknesses

---

> ### Author Response · Authors · 2025-11-15
>
> Thank you for your positive evaluation of our work! **We have revised the manuscript in light of your review, with all changes highlighted in blue**. Below are our responses to your questions:
> 1. *This paper does not account for semantic-level robustness. (**Appendix F.3 in Revision**)*
> - We appreciate the suggestion! Building on the inference results in Table 2, we have added a complementary experiment based on semantic similarity in **Appendix F.3**.
>
> | Model       | MATH Semantic | MATH OF | MMLU-Pro Semantic | MMLU-Pro OF | GPQA Semantic | GPQA OF |
> | --- | --- | --- | --- | --- | --- | --- |
> | Llama2-7b   | 0.851   | 0.475   | 0.706       | 0.622       | 0.760   | 0.509   |
> | Llama3.1-8b | 0.777   | 0.366   | 0.631       | 0.350       | 0.667   | 0.467   |
> | Llama-R1-8b | 0.669   | 0.158   | 0.601       | 0.292       | 0.575   | 0.371   |
> | Qwen3-8b    | 0.653   | 0.097   | 0.579       | 0.162       | 0.554   | 0.214   |
>
> - The experimental results are shown above. From this, we can observe the same trend as in Table 2, indicating that the semantic-level fluctuation mirrors the behavior captured by the OF metric and thereby supports our main conclusion.
>
> 2. *This paper does not specify the exact prompt templates used in the experiments. (**Appendix E.2 in Revision**)*
> - As discussed in Lines 306-308, our base prompt **follows the Llama-3.1 family**, which can be found in [this url](https://huggingface.co/datasets/meta-llama/Llama-3.1-8B-Instruct-evals), **we also add the prompts in the revision in Appendix E.2**. To simulate input perturbations, we synthesize additional prompts using three prompt-optimization methods and evaluate them across these prompts to induce controlled input variability (the three methods correspond to mainstream prompt optimizers used in our experiments).
>
> 3. *Given that different models vary in embedding initialization and pretraining corpora, how generalizable is this finding across architectures?*
> - Our analysis targets the general Transformer paradigm rather than a single model series. Thus, it is expected to generalize. Differences in embedding initialization and pre-training corpora can change the magnitudes of input and hidden-state vector norms, which in turn modulate robustness across models. Empirically, we observe consistent patterns in Llama-2, Llama-3.1, Llama-R1-Distilled, and Qwen-3, which corroborate the architectural generality of our conclusion.

---

> ### Author Response · Authors · 2025-11-28
>
> Dear Reviewer, we have revised the paper based on your review and submitted a response, and we sincerely hope that our changes address your concerns. As there are now fewer than five days remaining before the rebuttal period ends, we would be truly grateful for your feedback and sincerely look forward to your reply.

---

### Author Response · Authors · 2025-11-28
**Rebuttal Summary for AC**

Dear AC,

Thank you very much for your contribution to the conference! To help reduce your workload, we briefly summarize below the main content of our paper, the key reviewer concerns, and our responses.

Before the leakage on Nov 27,2025:
* Reviewer Kkzt increased score $(4\rightarrow 6)$ on Nov 26th.
* Reviewer K3vW maintained the positive score $(6 \rightarrow 6)$ on Nov 27th.

The final average score is **5.5 (6/6/4/6)**.

----

Our paper theoretically studies the robustness of CoT reasoning under input perturbations. We model the reasoning process as a bivariate Lipschitz function and derive an upper bound on output fluctuation. The main conclusions are:

1. As the number of CoT steps increases, the radius of tolerable input perturbations grows, but **even when the number of reasoning steps tends to infinity, the impact of input perturbations cannot be completely removed**.
2. For the Linear Self-Attention (LSA) model, the certified perturbation radius is **negatively correlated with the norms of the input embedding and hidden-state vectors**.

----

The strengths acknowledged by the reviewers:
1. Provide a formal theoretical derivation of the robustness and bounds of CoT reasoning. (All Reviewers)
2. Estabilish links between perturbations and reasoning steps / embed norms. (Reviewers 4uQH, Kkzt)
3. Provide insightful and informative analysis of Linear Self-Attention / Transformer. (Reviewers 4uQH, Kkzt)
4. Conduct comprehensive empirical validation across various models and datasets. (Reviewers 4uQH, Kkzt)

---

The main reviewer concerns and our corresponding responses are as follows:

1. **Using only OF as the robustness metric, lack of semantic-level analysis (M3bE)**
* In the revision, we add a **semantic similarity–based robustness experiment (“Semantic”)** in the appendix. The results show trends consistent with OF, indicating that OF is a reasonable proxy for semantic-level output fluctuation.

2. **Unclear prompt templates and perturbation procedure (M3bE, Kkzt)**
* In the revised version, we specify the exact prompt templates used. We also clarify that input perturbations are generated by applying existing prompt optimization methods (TextGrad, OPRO, CFPO) to rewrite prompts, thereby inducing changes in the embedding and hidden-state norms.

3. **Lipschitz assumption, bivariate modeling, and LSA simplification may be too strong (K3vW, 4uQH, Kkzt)**
* We point out that the Lipschitz assumption is a **standard assumption in recent theoretical work on Transformers**, and we refer to empirical validation in the appendix. Regarding the concern that the input consists of more than two tokens, we clarify that (x) and (h) denote the full semantic vector of the query and the hidden state, respectively. In the appendix, we further extend the analysis to the multi-token setting and show that the conclusions remain unchanged. For the LSA simplification, the revised version includes a non-linear analysis of standard softmax attention in the appendix, showing that non-linearity mainly changes the constants but not the structure of the robustness bounds. Experiments on real LLMs (Llama, Qwen) exhibit trends consistent with the theoretical predictions.

4. **Lack of intuitive and empirical support for the negative correlation between embedding/hidden-state norms and robustness (4uQH, Kkzt)**
* In the appendix, we discuss why measuring **absolute** output perturbation, rather than relative perturbation, better matches the perspective of decision flips (logit changes). In addition, we provide per-dataset correlation analyses between norms and OF (Pearson correlation coefficients around 0.4–0.6). Overall, larger norms are associated with higher OF, which empirically supports the theoretical negative correlation between norms and robustness.

5. **EM decreases as CoT length increases—does this undermine the claim that more steps improve robustness? (Kkzt)**
* Our additional analysis shows that as the number of CoT steps increases, performance usually **first increases and then decreases**; very long CoT chains mostly occur on harder problems, so the EM drop mainly reflects problem difficulty. At the same time, OF decreases monotonically with CoT length, indicating that even when answers are wrong, outputs under different prompts become more consistent (“consistently wrong”). From the robustness perspective, this still supports the conclusion that longer reasoning chains reduce output fluctuation.

6. **Limited methodological contribution, and issues of code and reproducibility (4uQH)**
* We clarify that our main focus lies in theoretical analysis and empirical validation. And our prompt selection method is intended as an **application and validation of the theoretical robustness bounds**, rather than to present a new standalone method. The revised version provides an anonymous code repository link for reproducibility.

Thank you again for your time and effort.

Best Regards,

Authors

---

### Meta-Review · Area_Chair_pZfo · 2025-12-23

**Summary:**

The paper presents a theoretical analysis of the robustness of Chain-of-Thought (CoT) reasoning in Large Language Models. By modeling the reasoning process as a bivariate Lipschitz function and utilizing Linear Self-Attention (LSA) as a proxy for Transformers, the authors derive theoretical bounds on output fluctuations under input perturbations.
The key findings are that robustness improves with the number of reasoning steps (though never perfectly) and is negatively correlated with the norms of input embeddings and hidden states. Reviewers generally appreciated the formal theoretical grounding and the comprehensive empirical validation across multiple models (Llama, Qwen) and datasets. Primary concerns revolved around the simplifying assumptions (LSA, Lipschitz continuity), the validity of the "Output Fluctuation" metric compared to semantic meaning, and the counter-intuitive nature of the findings regarding vector norms.

**Reviewer Concerns:**

**Addressed:**

1. Simplifying Assumptions (LSA/Lipschitz): Reviewers 4uQH and Kkzt were concerned that Linear Self-Attention was too simple a proxy. The authors addressed this by adding a derivation for standard non-linear softmax attention in the appendix, showing the qualitative bounds hold.

2. Robustness Metric (Reviewer M3bE): The concern that "Output Fluctuation" (OF) might not capture semantic meaning was addressed by adding new experiments using semantic similarity metrics, which yielded consistent trends.

3. Counter-intuitive Norm Results (Reviewer Kkzt): The reviewer questioned why larger norms reduce robustness. The authors clarified that absolute perturbation bounds are the relevant factor for decision flips in classification, validating the negative correlation.

4. Code/Reproducibility (Reviewer 4uQH): The authors provided an anonymized code repository.

5. Experimental Details (Reviewers M3bE, Kkzt): The authors clarified the prompt rewriting procedures and templates used for perturbations.

**Outstanding:**

1. Methodological Novelty (Reviewer 4uQH): The reviewer felt the prompt optimization method was auxiliary. While the authors argued it serves as validation for the theory, the core contribution remains theoretical rather than a new SOTA engineering method.

2. Generalizability of Assumptions (Reviewer K3vW): While the authors provided justifications and empirical backing, some skepticism regarding the strict adherence of real-world LLMs to the Lipschitz assumptions may remain, though the reviewer maintained a positive score.

**Reviewer Scores:**

Reviewer M3bE: 6 (maintained)

Reviewer K3vW: 6 (explicitly maintained; acknowledged clarifications).

Reviewer 4uQH: 4 (maintained).

Reviewer Kkzt: 4 -> 6 (explicitly stated).

---

### Decision · Program_Chairs · 2026-01-26

Accept (Poster)